# Indications for particle precipitation impact on the ion-neutral collision frequency analyzed with EISCAT measurements

Florian Günzkofer[1], Gunter Stober[2,3], Johan Kero[4], David R. Themens[5,6], Anders Tjulin[7], Njål Gulbrandsen[8], Masaki Tsutsumi[9,10], and Claudia Borries[1]

[1]Institute for Solar-Terrestrial Physics, German Aerospace Center (DLR), Neustrelitz, Germany
[2]Institute of Applied Physics, Microwave Physics, University of Bern, Bern, Switzerland
[3]Oeschger Center for Climate Change Research, Microwave Physics, University of Bern, Bern, Switzerland
[4]Swedish Institute of Space Physics (IRF), Kiruna, Sweden
[5]Space Environment and Radio Engineering Group (SERENE), School of Engineering, University of Birmingham, B15 2TT Birmingham, UK
[6]Department of Physics, University of New Brunswick, 8 Bailey Drive, PO Box 4440, Fredericton, NB E3B 5A3, Canada
[7]EISCAT AB, Kiruna, Sweden
[8]Tromsø Geophysical Observatory, UiT - The Arctic University of Norway, Tromsø, Norway
[9]National Institute of Polar Research, Tachikawa, Japan
[10]The Graduate University for Advanced Studies (SOKENDAI), Tokyo, Japan

**Correspondence:** Florian Günzkofer (florian.guenzkofer@dlr.de)

**Abstract.** The ion-neutral collision frequency is a key parameter for the coupling of the neutral atmosphere and the ionosphere. Especially in the mesosphere lower-thermosphere (MLT), the collision frequency is crucial for multiple processes e.g. Joule heating, neutral dynamo effects, and momentum transfer due to ion drag. Few approaches exist to directly infer ion-neutral collision frequency measurements in that altitude range. We apply the recently demonstrated difference spectrum fitting method to obtain the ion-neutral collision frequency from dual-frequency measurements with the EISCAT incoherent scatter radars in Tromsø. A 60-hour-long EISCAT campaign was conducted in December 2022. Strong variations of nighttime ionization rates were observed with electron densities at 95 km altitude varying from $N_{e,95} \sim 10^9 - 10^{11}$ m$^{-3}$ which indicates varying levels of particle precipitation. A second EISCAT campaign was conducted on 16 May 2024 capturing a Solar Energetic Particle (SEP) event, exhibiting constantly increased ionization due to particle precipitation in the lower E region $N_{e,95} \gtrsim 5 \cdot 10^{10}$ m$^{-3}$. We demonstrate variations of the ion-neutral collision frequency profile that we interpret as neutral particle uplift due to particle precipitation heating. Assuming a rigid-sphere particle model, we derive neutral density profiles which indicate a significant variation of neutral gas density between about 90 - 110 km altitude that correlate with the estimated strength of particle precipitation. However, the change of ion-neutral collision frequencies cannot be conclusively linked to the particle precipitation impact, and alternative interpretations are discussed. We additionally test the sensitivity of the difference spectrum method to various *a priori* collision frequency profiles.

# 1 Introduction

The neutral atmosphere dynamics in the mesosphere lower-thermosphere (MLT) region are affected by the lower atmospheric wave-driven dynamics and the forcing due to space weather (Liu, 2016). Therefore, this region is significant for atmosphere-ionosphere coupling and consequently the impact of space weather on the Earth system including the middle and lower atmosphere. Although the neutral particle density in the MLT region can only be measured *in situ*, it still is possible to infer the ion-neutral collision frequency $\nu_{in}$ from remote sensing measurements. The ion-neutral collision frequency is directly correlated to the particle density of the neutral atmosphere $n_n$. Assuming rigid-sphere collisions, the ion-neutral collision frequency is given by Chapman (1956)

$$\nu_{in} = 2.6 \cdot 10^{-9} \cdot \left( n_n \left[ \mathrm{cm}^{-3} \right] + n_i \left[ \mathrm{cm}^{-3} \right] \right) \cdot A^{-0.5}. \tag{1}$$

Here, the mean molecular ion mass $A$ is given in atomic mass units. Equation 1 assumes that the density of the neutral atmosphere is significantly larger than the ion density $n_i$ (which is assumed to be equal to the electron density $n_e$). There are several alternative ways to describe ion-neutral collisions, e.g. Maxwell collisions of ions and polarized neutrals (Schunk and Walker, 1971). Additionally, resonant collisions of neutrals with their first positive ion (e.g. $O_2$ and $O_2^+$) strongly increase the total collision frequency above certain temperature thresholds (Ieda, 2020). In this paper, Equation 1 is applied to relate the ion-neutral collision frequency, and the neutral density and potential deviations is discussed in Section 5.

$\nu_{in}$ is known to impact the shape of spectrum for incoherent scatter radar (ISR) measurements (Grassmann, 1993a; Akbari et al., 2017). Previous studies have demonstrated that the ion-neutral collision frequency can be obtained from dual-frequency ISR measurements (Grassmann, 1993b; Nicolls et al., 2014; Günzkofer et al., 2023b). However, dual-frequency ISR measurements are, at the moment, only possible with the EISCAT ultra-high and very-high frequency (UHF and VHF) radars. Therefore, the total number of dual-frequency ISR measurements remains sparse. Additionally, the multi-parameter analysis for two ISR spectra as proposed by Nicolls et al. (2014) is not part of the standard ISR analysis software. The *difference spectrum* fitting described in Grassmann (1993b) and demonstrated by Günzkofer et al. (2023b) does overcome this problem by combining the two spectra after the standard single-frequency analysis. Although the difference spectrum method has been known for several decades, a systematic application of the technique is still missing, and, thus, the ion-neutral collision frequency and neutral density in the MLT region have not been studied extensively leveraging dual-frequency EISCAT observations.

One forcing mechanism specifically important at high latitudes is the precipitation of energetic particles along the magnetic field lines down to MLT altitudes. These particles contribute significantly to the ionization and the heating of the high-latitude thermosphere. In the MLT region, mainly precipitating electrons with energies of $10 - 100$ keV and protons with energies of about 1 MeV contribute to the ionization of the atmosphere (Fang et al., 2010, 2013). Additionally, it has been shown that the heating due to the absorption of (extreme-) ultraviolet radiation and Joule heating alone is not sufficient to explain the observed thermosphere dynamics (Smith et al., 1982). Thermospheric heating leads to an up-welling of the neutral atmosphere and therefore causes distinct increases of the neutral particle density, and consequently also the ion-neutral collision frequency $\nu_{in}$, at certain altitudes (Hays et al., 1973; Olson and Moe, 1974; Kurihara et al., 2009; Oyama et al., 2012). The additional

ionization due to particle precipitation also increases the ionospheric conductivity and thereby the Joule heating (Vickrey et al., 1982). Both the direct particle precipitation heating and the additional Joule heating contribute significantly to the generation of ionospheric irregularities, e.g. large-scale traveling ionospheric disturbances (Sheng et al., 2020; Nykiel et al., 2024). It can be seen that the particle precipitation on the MLT region plays a crucial role in space weather research and the development of thermosphere-ionosphere models (Zhang et al., 2019; Watson et al., 2021).

In this study, we investigate the impact of particle precipitation on the vertical profiles of ion-neutral collision frequency and neutral particle density in the MLT region. The ion-neutral collision frequency is inferred from combined EISCAT UHF and VHF measurements. The particle precipitation impact can be estimated from EISCAT electron density measurements. The measurement campaigns with the EISCAT ISRs are described in Section 2. The difference spectrum method applied to determine ion-neutral collision frequencies and the estimation of the particle precipitation energy impact is described in Section 3. The obtained results are presented in Section 4. In Section 5, it is discussed how other processes like atmospheric tides and Joule heating might contribute to the observed variation of the ion-neutral collision frequency. Additionally, the low electron densities in the MLT region lead to considerable uncertainties in ISR measurements. The potential issues of increased data noise on the difference spectrum method are discussed in Section 5. The paper is concluded in Section 6 including an outlook on potential future work.

## 2    EISCAT UHF and VHF measurements

Dual-frequency ISR measurements can be performed with the Ultra-High Frequency (UHF) and the Very-High Frequency (VHF) radars near Tromsø, Norway (69.6° N, 19.2° E) operated by the EISCAT Scientific Association (Folkestad et al., 1983). The UHF ISR applies a radar frequency of 929 MHz with a nominal power of about $1.5-2$ MW and the VHF ISR transmits on a radar frequency of 224 MHz and has a nominal power of about 1.5 MW. The dual-frequency analysis requires both systems to be operated in the same radar mode and beam pointing to ensure overlapping observation volumes. A summary of all EISCAT instruments and experimental modes can be found in Tjulin (2024).

In this study, we leverage a 60-hour-long dual-frequency EISCAT campaign conducted from 13 December 2022 00 UT to 15 December 2022 12 UT during the Geminid meteor shower. A second EISCAT campaign, conducted on 16 May 2024 at 06 - 15 UT, is analyzed as well. This campaign was scheduled to be conducted during a Solar Energetic Particle (SEP) event and therefore exhibits high particle precipitation rates. The strength of the auroral electrojet during both measurement campaigns is estimated from the SuperMAG SME index (Newell and Gjerloev, 2011; Gjerloev, 2012) shown in Figure 1.

During both campaigns, the UHF and VHF radars were pointed in the zenith using the manda pulse code, which is optimized for high-resolution D-region measurements (as used in the *EISCAT Common Programme 6*). By default, *manda* measurements are analyzed in very narrow range gates with an altitude resolution of a few hundred meters up to 110 km altitude. However, for small electron densities, this results in very noisy data which causes problems with both the standard plasma parameter analysis and the dual-frequency analysis of collision frequencies. Therefore, we adjusted the altitude gates to 250 logarithmically-spaced gates from 50 - 200 km altitude. In the MLT region, this results in an altitude resolution of approximately 3 km. For the same

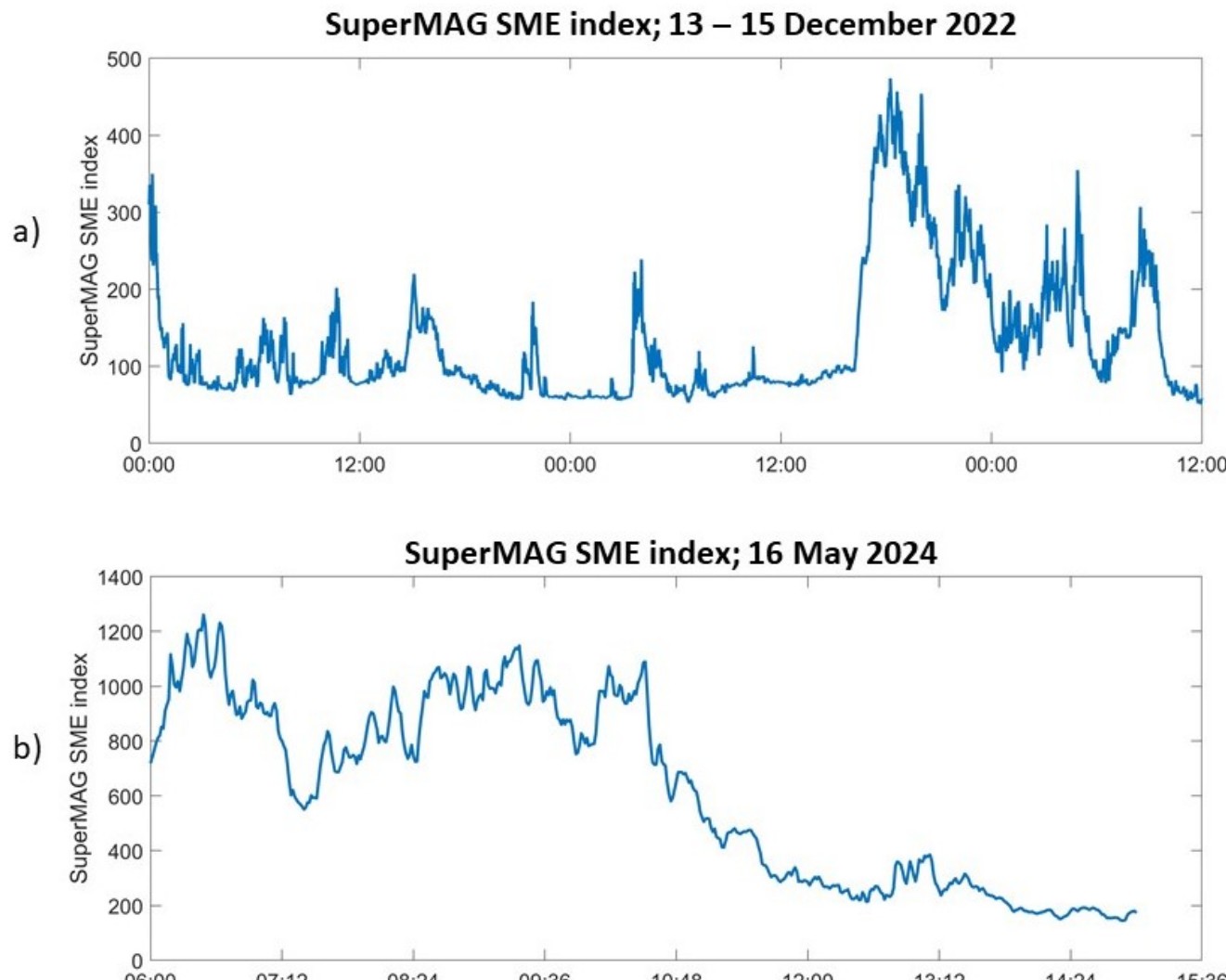

**Figure 1.** SuperMAG auroral electrojet index SME during the EISCAT campaigns in a) December 2022 and b) May 2024.

reason, the commonly applied integration window length of 60 s was extended to 120 s. The potential issues arising with too narrow altitude gates and too short integration windows are discussed in Section 5. The standard analysis software for EISCAT ISR measurements is the Grand Unified Incoherent Scatter Design and Analysis Package (GUISDAP) (Lehtinen and Huuskonen, 1996). For the analysis presented in this paper, the GUISDAP Version 9.2 was applied.

## 3 Methods

### 3.1 Difference spectrum fitting

The difference spectrum fitting is one of three methods to obtain ion-neutral collision frequencies from dual-frequency ISR measurements proposed by Grassmann (1993b). It was applied in Günzkofer et al. (2023b) where a detailed description of the method is given. The main advantage of the difference spectrum fitting is that it is based on the standard EISCAT ISR analysis package GUISDAP. Therefore, the implementation of specific software for the joint analysis of two ISR measurements as described in Nicolls et al. (2014) is not required.

In the first step, the UHF and VHF measurements are analyzed separately and in the second step, the obtained ISR spectra are combined. In this second step, the measured VHF spectrum is scaled to UHF frequencies with the UHF-to-VHF frequency ratio $\xi \approx 4.15$. The scaled VHF spectrum is equivalent to a UHF spectrum for an electron density $\xi^2 \cdot n_e$ and an ion-neutral collision frequency $\xi \cdot \nu_{in}$. Hence, the collision frequency $\nu_{in}$ is inferred from the difference between UHF and scaled VHF spectra. Technical differences between the two radars are accounted for by introducing the so-called $\beta$ parameter which is determined from the measurements at the uppermost range gate corresponding to approximately 200 km altitude. At this height, we assume a collision-less ionosphere, i.e. $\nu_{in} \ll \omega_i$ with the ion-gyrofrequency $\omega_i$ and, thus, the remaining differences are most likely given by system-specific factors such as beam width and differences in the observation volume. A detailed description of this procedure and the impact of varying $\beta$ parameters is outlined in Günzkofer et al. (2023b), Section 3 and 4.

### 3.2 Particle precipitation estimate

Precipitating particles affect the MLT by ionizing the neutral molecules in that region. The ionization electrons are thermalized and thereby heat both the ionosphere plasma and the MLT neutral atmosphere. Assuming particle precipitation to be the dominant ionization process, the energy deposition can be estimated from ISR electron density measurements. Vickrey et al. (1982) demonstrated a method to determine the particle precipitation energy deposition assuming an empirical profile for the effective recombination coefficient. However, it has been shown that the effective recombination coefficient profile depends on the precipitating particles (electrons or protons) and energy (Gledhill, 1986). As an approximate quantification for the total particle precipitation impact, the electron density at 95 km altitude $N_{e,95}$ measured with the EISCAT VHF ISR is applied. The validity of this approximation and possible problems are discussed in Section 5.

## 4 Results

### 4.1 EISCAT Geminids campaign December 2022

As described in Section 2, a 60-hour dual-frequency EISCAT campaign from December 2022 is analyzed. Figure 2 shows the measured electron density and the ion-neutral collision frequency calculated with the difference spectrum method. Both quantities are shown at 85 - 110 km altitude where we expect particle precipitation to have the strongest impact. At higher

altitudes, the Joule heating would become more and more significant (see Section 5) (Baloukidis et al., 2023; Günzkofer et al., 2024). The electron density at 95 km altitude $N_{e,95}$ that is applied as a quantification for the particle precipitation impact is shown as well. Since the *manda* experiment mode is optimized for the EISCAT VHF radar, the electron density is taken from these measurements. The time axis in Figure 2 is given in Universal Time (UT). The local apparent solar time (LAST) at Tromsø ($\sim 20°$ E) is approximately UT + 80 min.

It can be seen in Figure 2 a) that the electron density is significantly increased at nighttime. At high latitudes, it can be assumed that particle precipitation is the dominant source of nighttime ionization. During the last night from 14 to 15 December, the increase in electron density is much stronger than in the two nights before, indicating strong particle precipitation presumably due to substorm activity (see Figure 1). The electron density at 95 km altitude in Figure 2 c) shows maxima at about 13 December 00:40 UT and 22:00 UT, as well as from 14 December 19:00 UT to 15 December 04:00 UT. At the times of high $N_{e,95}$, Figure 2 b) shows that the ion-neutral collision frequency is strongly increased at altitudes $\gtrsim$ 95 km. This suggests an effect of particle precipitation on the ion-neutral collision frequency.

To investigate the daily mean variation of the ion-neutral collision frequency, we bin the measured profiles in four LAST sectors (midnight, dawn, noon, and dusk). Figure 3 shows the median profiles for the four bins and two climatology profiles. The first climatology profile is applied as *a priori* profile for the EISCAT ISR analysis. In this case, the *a priori* climatology is taken from the CIRA2014 neutral atmosphere model (Rees et al., 2013). The applied climatology model, however, depends on the GUISDAP version and installation. The second climatology profile is calculated from the empirical NRLMSIS 2.0 model neutral densities (Emmert et al., 2021). As already seen in Günzkofer et al. (2023b), the two climatologies are different by a factor $\sim 1.5 - 2$ but both show a smooth exponential decrease of the collision frequency with increasing altitude. So in addition to the LAST binning, Figure 3 shows ion-neutral collision frequency profiles derived from EISCAT initializing the difference spectrum fit with a) the EISCAT *a priori* and b) the NRLMSIS climatological profile, respectively.

Below 100 km altitude the difference spectrum $\nu_{in}$ profile appears to be strongly affected by the choice of *a priori* profile. However, it can be seen in Figure 3 a) that above 95 km, the difference spectrum fit starts to deviate from the EISCAT *a priori* profile towards the NRLMSIS profile. This agrees well with the results found in Günzkofer et al. (2023b). In Section 5, it is discussed at which altitude the difference spectrum fit is *a priori dominated*.

Above $100 - 105$ km and with increasing electron density, our fitting approach starts to get more and more independent of the choice of the *a priori* profile, which is indicative of a sufficient measurement response. At the highest investigated altitude of 110 km, the four profiles give very similar values in both plots. Also the daily variation of $\nu_{in}$ above 100 km altitude is identical in Figure 3 a) and b). It can be seen that the ion-neutral collision frequency is significantly increased during the LAST midnight and dawn sectors. The lowest $\nu_{in}$ values are found during the LAST noon and dusk sectors.

The diurnal variation found in Figure 3 fits the impact of particle precipitation, especially the dawn-dusk asymmetry as the particle precipitation energy deposition is known to be larger around the morning hours (Vickrey et al., 1982). In the next step, we separate the vertical profiles of the ion-neutral collision frequency $\nu_{in}$ with respect to the particle precipitation impact, quantified by $N_{e,95}$. We define three ranges for low, medium, and high particle precipitation at $N_{e,95} < 1 \cdot 10^{10}$ m$^{-3}$, $1 \cdot 10^{10}$ m$^{-3} < N_{e,95} < 2 \cdot 10^{10}$ m$^{-3}$, and $N_{e,95} > 2 \cdot 10^{10}$ m$^{-3}$. For each bin, the median vertical $\nu_{in}$ profile is calculated. For

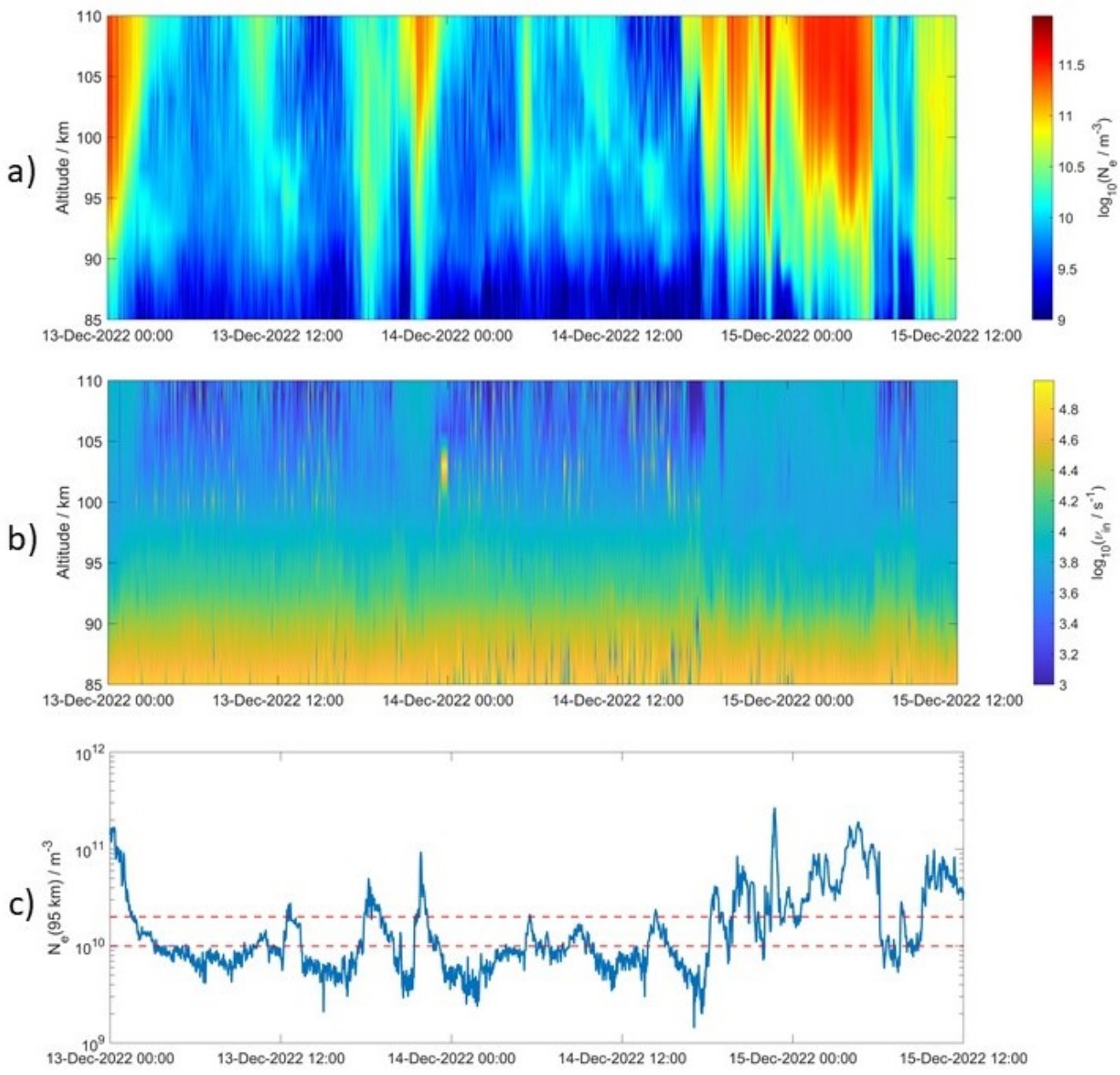

**Figure 2.** EISCAT measurements from 13 December 2022 00 UT to 15 December 2022 12 UT. a) EISCAT VHF electron density, b) ion-neutral collision frequency from combined VHF and UHF measurements, and c) $N_{e,95}$ from the VHF electron density.

this analysis, we only apply the collision frequencies obtained from the dual-frequency fit initialized with the NRLMSIS climatology.

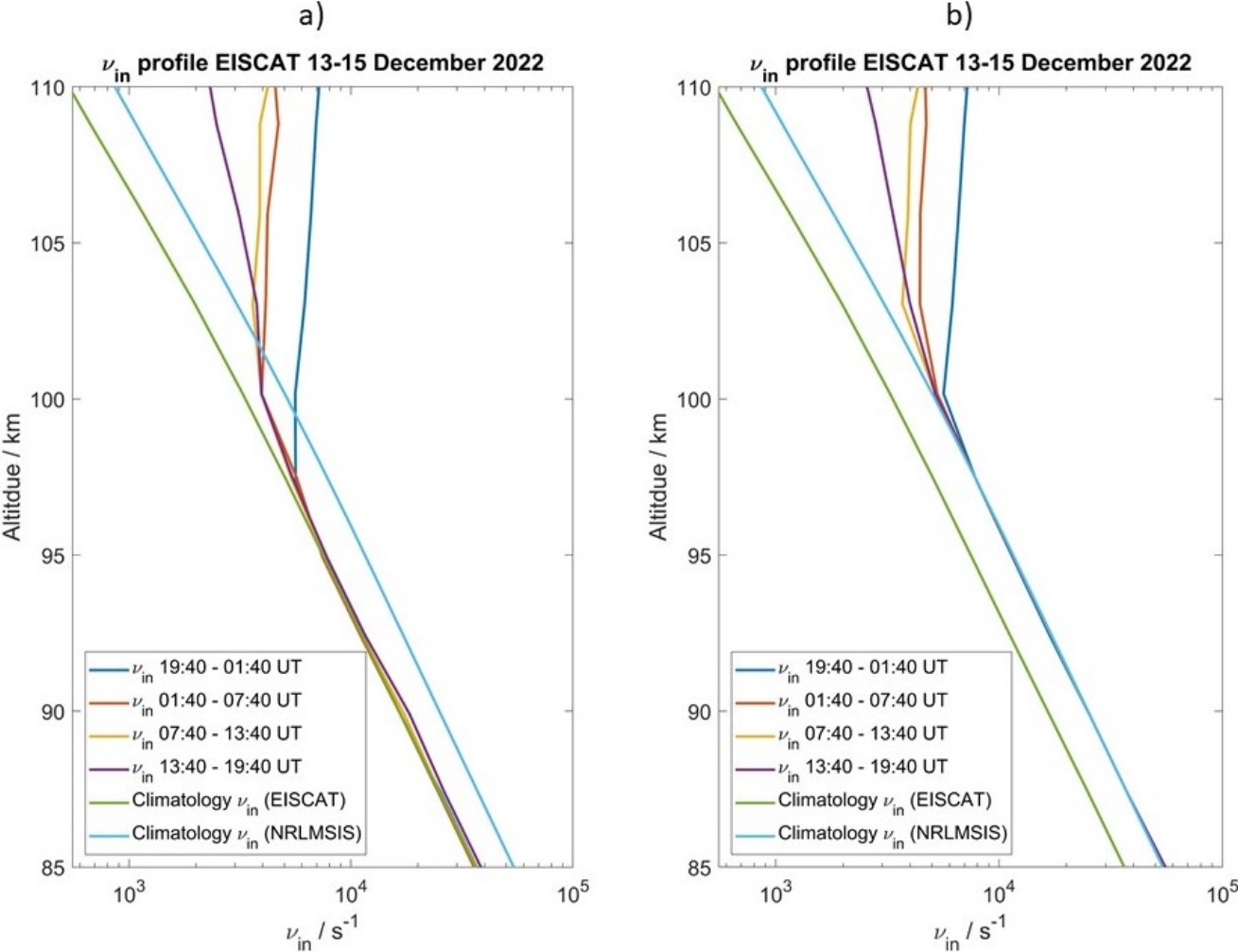

**Figure 3.** Median vertical profiles of the ion-neutral collision frequency for local apparent solar midnight, dawn, noon, and dusk sectors compared to climatology profiles. As *a priori* profile for the difference spectrum fit a) the EISCAT single-frequency $\nu_{in}$ or b) $\nu_{in}$ calculated from NRLMSIS results can be applied. The impact of the *a priori* on the difference spectrum fit is discussed in Section 5 and illustrated in Figure 9.

Figure 4 a) shows the median profiles for the three bins and the climatology profile. The interquartile is shown for the bins with lowest and highest $N_{e,95}$ profiles, indicating the volatility of the difference spectrum fit for low electron densities.

Dual-frequency $\nu_{in}$ measurements are highly unreliable for low ionization conditions. However, it still appears to be evident that $\nu_{in}$ increases with increasing particle precipitation energy deposition above about 100 km altitude which explains the daily variation found in Figure 3. Additionally, we found a characteristic decrease of the ion-neutral collision frequency for high $N_{e,95}$ at altitudes of about $90-100$ km. It can be seen, that for higher particle precipitation impact, the $\nu_{in}$ profiles

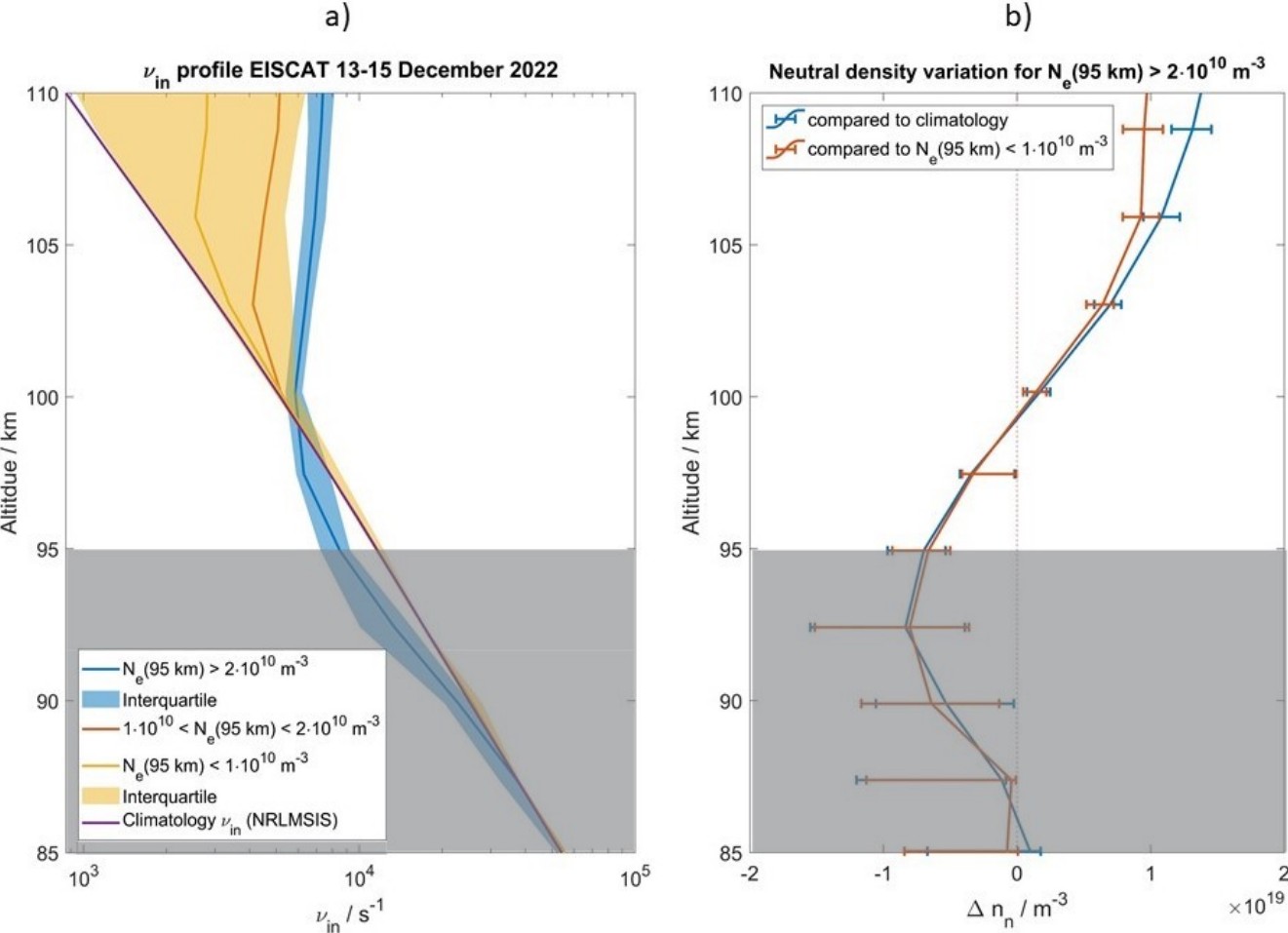

**Figure 4.** a) Median vertical collision frequency profiles binned with $N_{e,95}$. b) Difference in neutral particle densities calculated from the collision frequency profiles for high and low particle precipitation. The gray shaded areas indicate the altitudes at which the difference spectrum fit is *a priori* dominated (see Section 5).

deviate from the *a priori* profile at lower altitudes. For the lowest particle precipitation impact $N_{e,95} < 1 \cdot 10^{10}$ m$^{-3}$, the
collision frequency fit is dominated by the *a priori* profile up to 100 km altitude and mostly unreliable at higher altitudes. The statistical interquartile uncertainties of the high particle precipitation profile are shown as a blue shaded area in Figure 4 a). The gray shaded areas indicated the altitudes at which the difference spectrum fit is *a priori dominated* and the results cannot be considered reliable (see Section 5).

The decrease of $\nu_{in}$ at $\sim 90 - 100$ km and increase of $\nu_{in}$ at $\gtrsim 100$ km altitude with larger $N_{e,95}$ indicates an uplift of
neutral particles. Applying Equation 1, we can calculate the vertical profile of neutral particle density $n_n$ from the collision frequency profiles. Figure 4 b) shows the difference of the $n_n$ profiles obtained from the $\nu_{in}$ profiles for $N_{e,95} > 2 \cdot 10^{10}$ m$^{-3}$

compared to the low particle precipitation profile ($N_{e,95} < 1 \cdot 10^{10}$ m$^{-3}$) and the climatology profile. The depletion of the neutral particle density below 100 km is approximately equal to the increase above 100 km. In absolute numbers, the maximum $n_n$ decrease/increase are $\pm 10^{19}$ m$^{-3}$.

## 4.2 EISCAT SEP campaign May 2024

The dual-frequency EISCAT campaign conducted on 16 May 2024 was scheduled to be triggered by an SEP event. This allows us to study the impact of continuously high particle precipitation rates on the ion-neutral collision frequency over several hours. At approximately 100 km altitude, the main impact of particle precipitation is caused by Auroral electrons (Mironova et al., 2015). We focused our analysis on the campaign period with EISCAT measurements on 16 May 2024 from 06 - 15 UT. Figure 5 shows a) the electron density, b) the ion-neutral collision frequency, and c) $N_{e,95}$ on 16 May 2024.

The major difference compared to the first campaign is that the electron density in Figure 5 a) is generally larger than in Figure 2 a). This is presumably caused by the increased particle precipitation energy deposition and consequently increased ionization rate due to the SEP event. The increased electron density improves the SNR for all altitudes above 95 km. Since the solar zenith angle at the Tromsø geographic latitude is significantly lower in May compared to December, photoionization presumably contributes to the E region ionization. This is discussed in Section 5.

EISCAT measurements indicate that $N_{e,95} > 2 \cdot 10^{10}$ m$^{-3}$ for the entire campaign period on 16 May 2024. Therefore, all measurements on 16 May 2024 would fall in the highest $N_{e,95}$ range by which $\nu_{in}$ was sorted in Figure 4 a). The ion-neutral collision frequency measurements are binned with the electron density at 95 km altitude. Two bins are applied for $N_{e,95}$ values larger or smaller than $1 \cdot 10^{11}$ m$^{-3}$. Figure 6 a) shows the two $\nu_{in}$ profiles from 16 May 2024 in comparison to the NRLMSIS climatology profile and the $N_{e,95} > 2 \cdot 10^{10}$ m$^{-3}$ profile from December 2022.

For the May campaign we found that the two $\nu_{in}$ profiles for $N_{e,95} > 1 \cdot 10^{11}$ m$^{-3}$ and $N_{e,95} < 1 \cdot 10^{11}$ m$^{-3}$ are nearly identical and highly similar to the $N_{e,95} > 2 \cdot 10^{10}$ m$^{-3}$ profile from December 2022. Below about 90 km altitude, the seasonal variation of the NRLMSIS climatology that is used to initiate the dual-frequency $\nu_{in}$ fit causes the profiles to deviate. However, there are additional differences between the December 2022 and May 2024 profiles above 90 km altitude presumably caused by the difference in strength of particle precipitation. Equation 1 is applied to calculate the neutral particle density $n_n$ profiles from the $N_{e,95} > 1 \cdot 10^{11}$ m$^{-3}$ collision frequency profile. The difference in neutral particle density $\Delta n_n$ between the $N_{e,95} > 1 \cdot 10^{11}$ m$^{-3}$ and the climatology profile is calculated equivalent to the blue $\Delta n_n$ profile in Figure 4 b). Both $\Delta n_n$ profiles are shown in Figure 6 b).

It can be seen that the $\Delta n_n$ profile for May 2024 is shifted to lower altitudes by about 2 km compared to the December 2022 profile. Assuming particle precipitation to be the cause for the observed changes, this would mean that the deposition altitude is slightly lower during the May 2024 measurements, indicating a higher particle energy. However, the increase/decrease of neutral particles is of similar magnitude about $10^{19}$ m$^{-3}$.

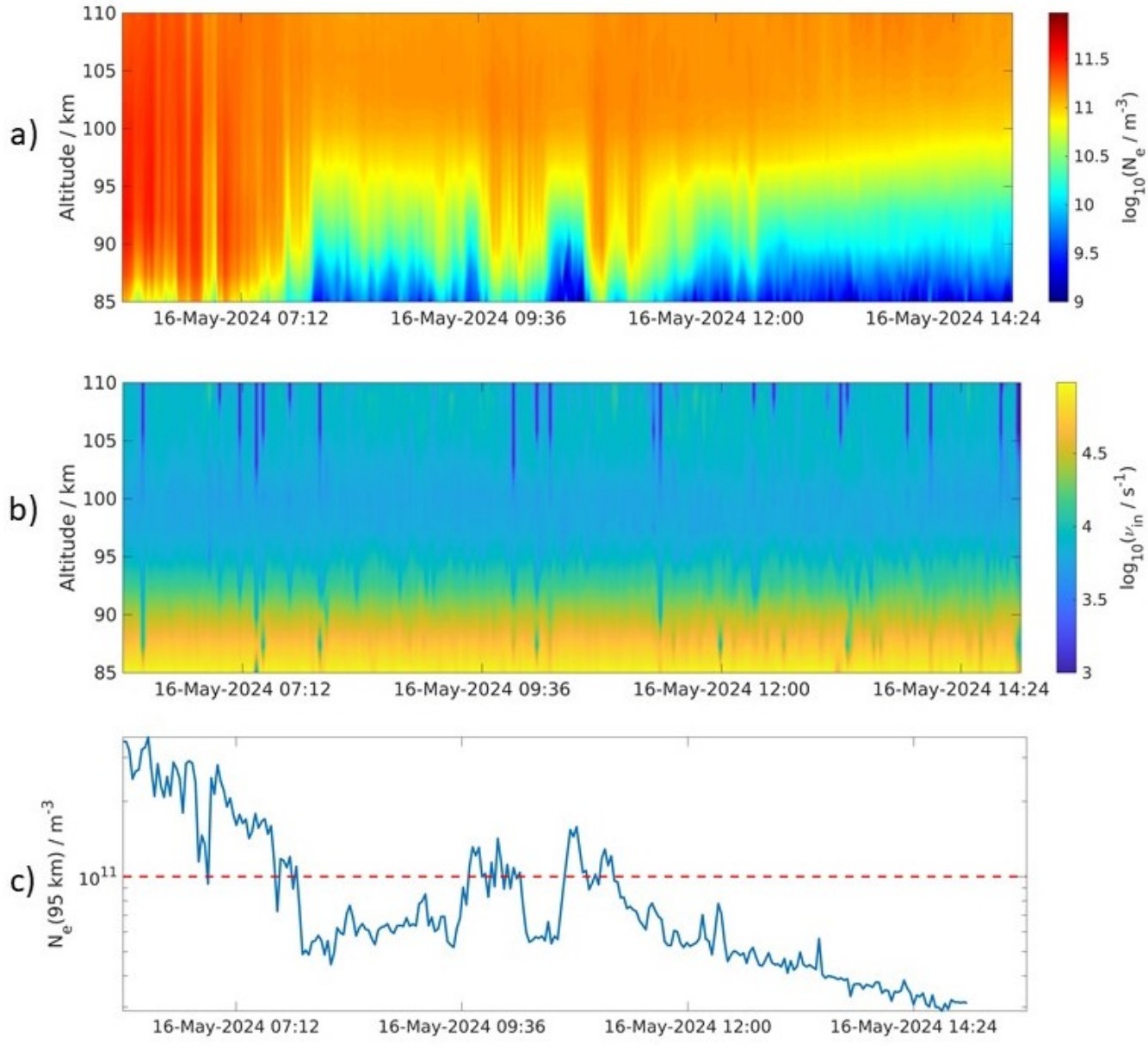

**Figure 5.** a) EISCAT VHF electron density, b) ion-neutral collision frequency from combined VHF and UHF measurements, and c) electron density at 95 km altitude from the VHF measurements.

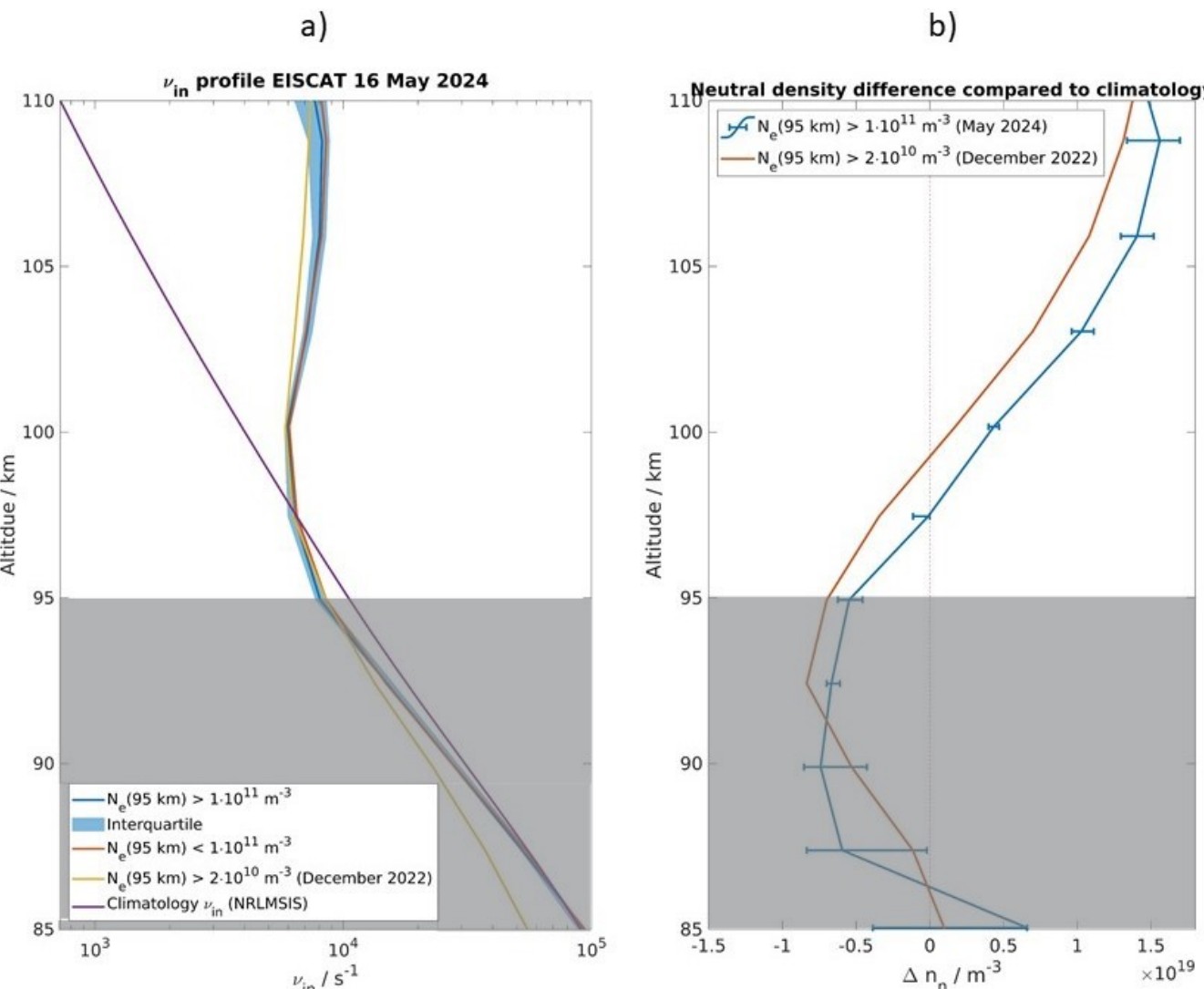

**Figure 6.** a) Median ion-neutral collision frequency profiles from 16 May 2024 binned for $N_{e,95}$ values larger or smaller than $1 \cdot 1^{11}$ m$^{-3}$, The $\nu_{in}$ profile from December 2022 for $N_{e,95} > 2 \cdot 10^{10}$ m$^{-3}$ is shown for comparison. b) Difference of the neutral density profiles calculated for $N_{e,95} > 1 \cdot 10^{11}$ m$^{-3}$ and $N_{e,95} > 2 \cdot 10^{10}$ m$^{-3}$ (December 2022) in comparison to the respective climatology profiles for May/December. The gray shaded areas indicate the altitudes at which the difference spectrum fit is *a priori* dominated (see Section 5).

## 5 Discussion

The presented work presumably shows the impact of particle precipitation on the ion-neutral collision frequency. However, the presented results underlie considerable uncertainties, both regarding the obtained results themselves as well as their interpretation. The following issues will be discussed in this Section:

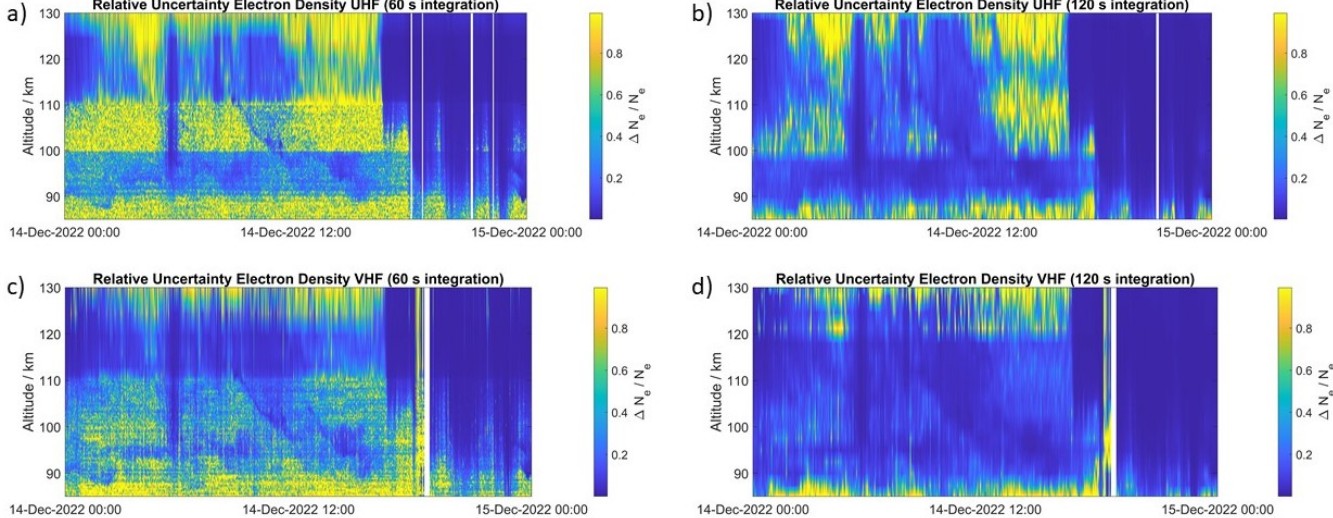

**Figure 7.** Relative uncertainty of electron density $\Delta N_e/N_e$ for EISCAT a), b) UHF and c), d) VHF measurements on December 14, 2022. Distinguished are a), c) the 'standard' case with 60 s integration windows and very high altitude resolution up to 110 km, and b), d) the analysis settings applied in this paper with 120 s integration and reduced altitude resolution.

- EISCAT data noise for low electron densities in the MLT region

- Impact of *a priori* parameter assumptions on the dual-frequency fit

- Energy balance and reaction time of neutral uplift

– Alternative explanations for neutral density variations (tilted isobars, atmospheric tides, and Joule heating)

- Calculation of the neutral densities and importance of resonant collisions

- Validity of the difference spectrum method

- Quantification of particle precipitation impact with $N_{e,95}$

## 5.1 EISCAT data noise

As mentioned in Section 2, the EISCAT single-frequency analysis was adjusted from the 'standard' case with 60 s integration windows and the typically high *manda* altitude resolution in the MLT region of a few hundred meters. For the results presented in Section 4, integration windows of 120 s and an altitude resolution of roughly 3 km in the MLT region were applied (see Section 2). Figure 7 illustrates the impact of these adjusted analysis settings on the data uncertainty.

    The relative electron density uncertainty $\Delta N_e/N_e$ for the 'standard' settings is shown in Figures 7 a) and c) for UHF 215   and VHF respectively. Both instruments exhibit considerable uncertainties below 90 km altitude, presumably due to the low electron density. The most prominent feature, however, is a band of strong noise between 100 km and 110 km altitude in the

UHF measurements. The lower boundary of this noise band can be explained with the settings of the ISR single-frequency analysis. Below 100 km, the ion temperature $T_i$ is not fitted but taken from an *a priori* model for UHF *manda* measurements. This limits the variability of the ISR fit and therefore reduces the uncertainty of the plasma parameters. Above 100 km, the ion temperature is also fitted; consequently, the uncertainty increases sharply above this altitude. The upper boundary of the noise band at 110 km is presumably caused by the narrow altitude gates in the standard *manda* analysis. Up to 110 km altitude, the standard analysis applies very narrow altitude gates of only a few hundred meters, resulting in a low SNR and the observed large uncertainties. Therefore, a sharp transition from high to low uncertainties at 110 km altitude can be observed in the VHF measurements as well.

For the adjusted measurement setting in Figures 7 b) and d) which have also been applied to obtain the results in Section 4, the uncertainties are notably lower. However, the UHF noise band above 100 km can still be seen at times of very low electron density (compare to Figure 2 a). Due to the adjusted altitude gates, the sharp transition at 110 km altitude is no longer present. For electron densities $N_e > 10^{10}$ m$^{-3}$, the uncertainties are reasonably small to be acceptable.

A potential issue of the high uncertainties when applying the 'standard' analysis settings is shown in Figure 8. During the December 2022 EISCAT measurements, the collision frequency profiles are binned with local apparent solar time equivalent to Figure 3 a). A notable jump is found in all profiles at 100 km altitude. This is presumably caused by the noise band shown in Figure 7 a). Due to the adjusted analysis settings, the kink has been entirely suppressed/removed in Figure 3.

The uncertainties for the May 2024 campaign are generally lower due to the increased electron densities and only become significant below 90 km altitude. However, applying the 'standard' altitude gates and a 60 s integration, a discontinuity of $\Delta N_e/N_e$ is found at 110 km altitude with a jump of about 50%. Due to the generally low relative uncertainties, this would not have affected the analysis strongly and the discontinuity disappeared when applying the adjusted analysis settings.

## 5.2   Impact of *a priori* collision frequency profile

*A priori* parameters are not only relevant for the single-frequency fit of ISR measurements but also for the dual-frequency method. To initialize the dual-frequency $\nu_{in}$ fit, an *a priori* collision frequency profile is applied. In Figure 3, the obtained collision frequencies for two different *a priori* profiles have been shown. It could be seen that at low altitudes (approximately below 95 km), the fitted profiles stick very closely to the *a priori* profiles. We can estimate the altitudes at which the difference spectrum fit is dominated by the *a priori* profile from the differences between the profiles in Figures 4 a) and 6 a) and the equivalently binned profiles obtained from fits with the EISCAT *a priori* profile. The profile differences for high and low $N_{e,95}$ profiles for December 2022 and the high $N_{e,95}$ profile for May 2024 are shown in Figure 9.

As the difference spectrum method involves a non-linear least-square fit of the difference in spectral amplitude, a low SNR value results in a low measurement response and the solution tends to stay much closer to the *a priori* profile. Although the absolute least-square errors are smaller when fitting the difference spectrum, the relative errors are much larger causing the fit to accept the *a priori* profile as the solution. It can be seen in Figure 9 that for the high $N_{e,95}$ $\nu_{in}$ profiles, the relative deviation of profiles obtained with different *a priori* profiles is considerably low above 100 km altitude. Therefore, the difference spectrum fit is not impacted by the choice of *a priori* profile there. Below 100 km altitude, the profiles start to deviate significantly

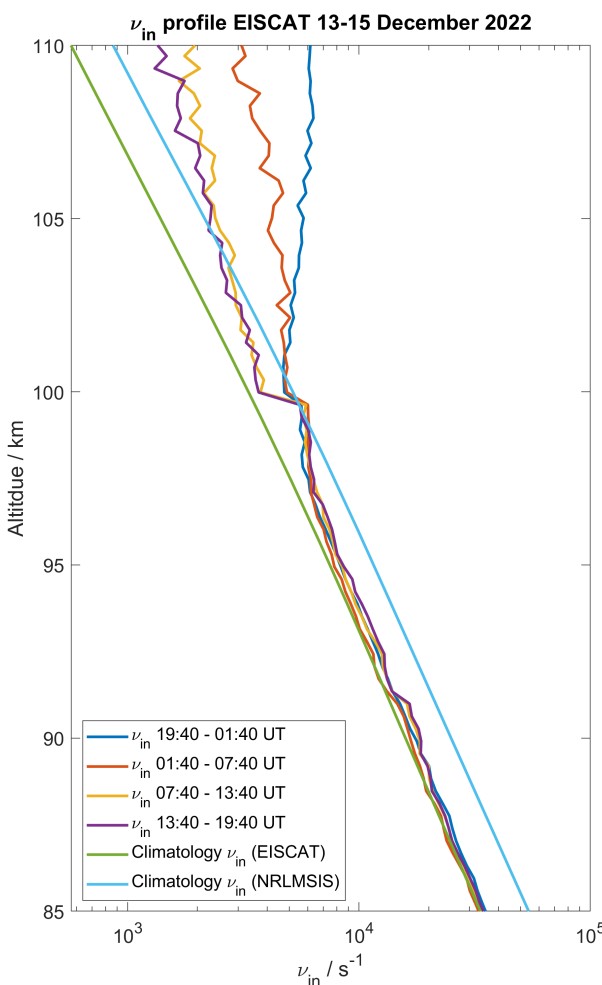

**Figure 8.** Equivalent to Figure 3 a) but for the 'standard' single-frequency EISCAT analysis settings.

and below about 95 km, the profile difference is nearly equivalent to the *a priori* profiles. Therefore, we determined that the difference spectrum fit is *a priori dominated* below 95 km altitude and the obtained $\nu_{in}$ profiles cannot be considered reliable. This altitude is therefore shown gray-shaded in Figures 4 and 6. The low $N_{e,95}$ profiles obtained during the December 2022 campaign show a considerable dependence on the choice of *a priori* profile at all altitudes. However, the profiles appear to be

not completely *a priori* dominated above about 105 km altitude.

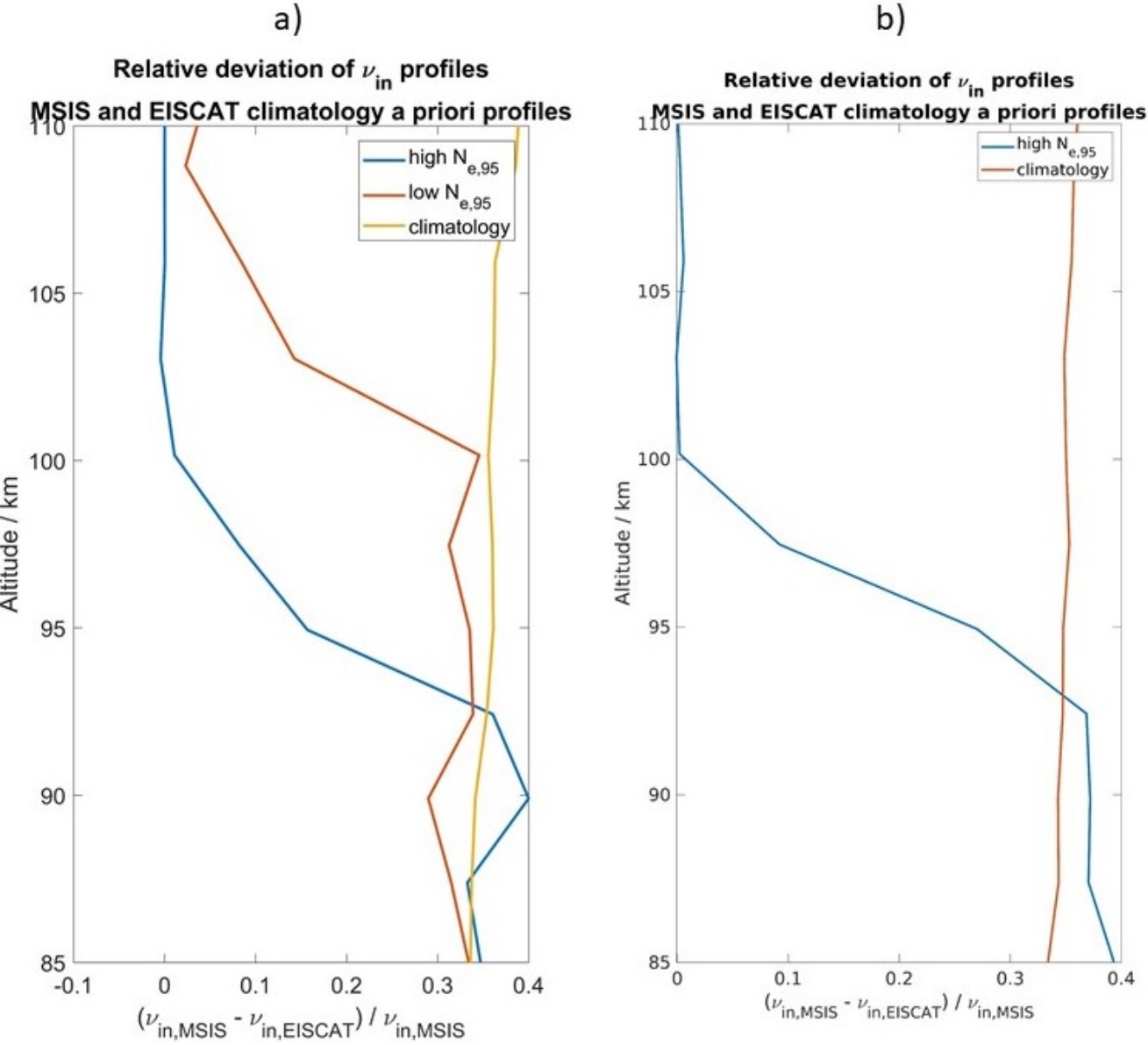

**Figure 9.** Deviation of $\nu_{in}$ profiles for *a priori* difference spectrum fit profiles from NRLMSIS or EISCAT climatology for a) December 2022 and b) May 2024.

### 5.3 Energy balance of neutral uplift

An estimation of the local energy deposition rate $q$ can be obtained from ISR electron density measurements applying the method described by Vickrey et al. (1982). This method applies a constant, empirical profile for the effective recombination rate obtained from various ionospheric and laboratory experiments. Though Gledhill (1986) showed that the effective recombination

rate depends on the dominant type of particle precipitation, the method described by Vickrey et al. (1982) is applied here to estimate the local heating rate at 95 km altitude. For $N_{e,95} > 2 \cdot 10^{10}$ m$^{-3}$, the local heating rate at 95 km altitude is $q > 0.88 \, \mu$W m$^{-3}$. The height-integrated energy deposition between 90 km and 110 km altitude reaches levels of approximately $Q \sim 3$ mW m$^{-2}$.

In order to assess the neutral uplift visible in the blue $\Delta n_n$ profile shown in Figure 4 b), we also calculate the energy density required for the uplift. We calculate the difference in total potential energy for the neutral particle density profiles $n_{n,1}$ (high $N_{e,95}$ conditions) and $n_{n,2}$ (low $N_{e,95}$ conditions)

$$\Delta E = \int_{90km}^{110km} (n_{n,1} - n_{n,2}) \cdot 29\mathrm{u} \cdot g \cdot h \, \mathrm{d}h \approx (1.12 \pm 2.92) \text{ kJ m}^{-2}. \tag{2}$$

This assumes a mean particle mass of 29 atomic mass units. At the above calculated height-integrated energy deposition rate, it would take approximately 100 h of particle precipitation to deposit the energy for the observed uplift. So for the presented measurements, it is unreasonable that the observed median uplift is caused by particle precipitation. However, the uncertainties in Figure 4 b) influence the calculation of the energy balance in Equation 2 quite significantly, causing energy uncertainties of $\pm 2.92$ kJ m$^{-2}$. The uncertainties are therefore far larger than the median energy difference calculated in Equation 2. This means that though the ion-neutral collision frequencies profiles in Figures 3, 4 a), and 6 a) can be inferred with reasonable uncertainty, the physical impact of these uncertainties is quite major. Therefore, a considerably higher accuracy of the difference spectrum $\nu_{in}$ measurements is required before quantitative implications can be drawn.

## 5.4 Reaction time of the atmosphere

Another point that needs to be considered is the reaction time of the atmosphere gas to the heating due to particle precipitation. For a long reaction time, the binning of $\nu_{in}$ profiles with $Q_P$ is not justified and the delay of the neutral uplift would need to be considered. This is especially important for the December 2022 measurements with strong fluctuations in the particle precipitation rate. We estimate the vertical neutral wind induced by the particle precipitation heating at 100 km altitude following Hays et al. (1973), Kurihara et al. (2009), and Oyama et al. (2012)

$$U_z = \frac{q}{\rho \left( c_p \frac{\delta T}{\delta z} + g \right)}. \tag{3}$$

The above estimated $q \sim 0.88 \, \mu$W m$^{-3}$ is applied here. The neutral mass density $\rho$, the specific heat capacity at constant pressure $c_p$, and the vertical neutral temperature gradient $\delta T/\delta z$ are obtained from the NRLMSIS model. This results in vertical winds of $U_z \sim 3.6$ m s$^{-1}$. The average vertical uplift of a particle can be estimated from the energy difference in Equation 2 as $\Delta h = \Delta E / (N_n \cdot 29\mathrm{u} \cdot g)$ with the height-integrated particle density $N_n \sim 3.75 \cdot 10^{23}$ m$^{-2}$. It should be noted that $N_n$ at 90 km to 110 km altitude is nearly equivalent for all neutral density profiles including the climatology NRLMSIS profile. Considering the large uncertainty of the energy difference calculation, the possible average uplift ranges from about 6 km to

19 km assuming the energy differences from Equation 2 larger than the median). The vertical velocity obtained from Equation 3 results in a reaction time of approximately $27-88$ min. Though the majority of $N_{e,95} > 2 \cdot 10^{10}$ m$^{-3}$ conditions during the December 2022 campaign occurred in one several hours-long interval during the night from 14 to 15 December, a reaction time $\sim 90$ min would definitely impact the analysis presented in this paper. Additionally, Grandin et al. (2024) showed that the typical duration of auroral precipitation events is around 20 min and therefore slightly lower than the calculated reaction time. However, Kurihara et al. (2009) noticed that the observed reaction time is usually significantly shorter than calculated from Equation 3. If the uplift is not caused by vertical winds but rather by a density wave, the disturbance would be transported with the wave's phase velocity, which would explain the shorter reaction time. However, the atmosphere reaction time to the particle precipitation generally needs to be considered when investigating short periods of strong particle precipitation. Due to the high uncertainty of the reaction time estimate it can not conclusively determined that the observed changes of the $\nu_{in}$ profile are caused by particle precipitation. Additionally, the lower typical duration of precipitation events suggests that other mechanisms contribute as well, even if particle precipitation is the main driver. It should also be noted that the vertical wind is an estimate and not directly observed, so all interpretations should be considered with care.

## 5.5 Vertical wind due to tilted isobars

Our measurements do not conclusively prove that the observed neutral density changes are caused by a neutral uplift due to particle precipitation heating. Since the atmospheric reaction time from the neutral uplift estimated in the previous section is presumably slightly too long, alternative explanations have to be considered.

Oyama et al. (2008) discussed the impact of tilted isobars due to localized heating events. For localized heating events, such as particle precipitation, the isobars given by the barometric formula are tilted with respect to geographic altitude. Hence, zonal and meridional winds along the isobars do have a geographically vertical component. Oyama et al. (2008) showed that this vertical wind component is generally in the order of a few ms$^{-1}$ but can be significantly larger for strong horizontal temperature gradients. Therefore, the vertical wind due to zonal and meridional winds along tilted isobars is at least of the same order as the uplift calculated in the previous section.

It should be noted that though the geomagnetic heating due to particle precipitation or Joule heating (see Section 5.7) causes the tilt of the isobar, vertical advection is not necessarily correlated to the particle precipitation observed with the EISCAT radar. In Oyama et al. (2008), the heating event took place 80 km from Tromsø, but if the horizontal wind direction over Tromsø is oriented toward the heating source, an upward vertical wind component is observed.

## 5.6 Atmospheric tides

Atmospheric tides are an important forcing mechanism of the MLT region (Lindzen, 1979; Becker, 2017). Neutral wind measurements from the Tromsø meteor radar were analyzed (Hall and Tsutsumi, 2013) to assess the tidal activity during the time of the above-described EISCAT campaigns. Technical details for this type of meteor radar can be found in Holdsworth et al. (2004). The Tromsø meteor radar is part of the Nordic Meteor Radar Cluster, which permits to obtain spatially resolved winds covering the same observation volume as EISCAT (Stober et al., 2021a; Günzkofer et al., 2023a). The meteor radar

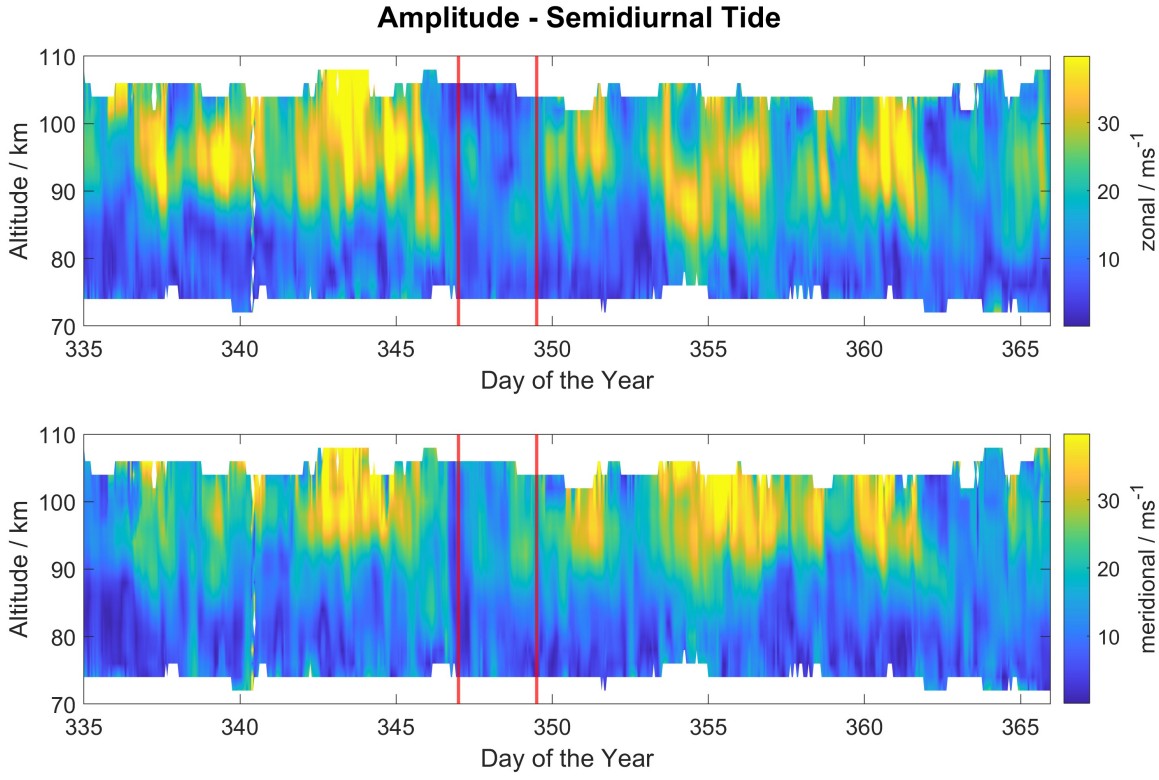

**Figure 10.** Amplitude of the semidiurnal tide in zonal (top) and meridional (bottom) neutral winds measured with the Tromsø meteor radar. The vertical red lines mark the beginning and the end of the EISCAT Geminids campaign during December 2022.

provides measurements of the neutral wind velocities at approximately $70 - 110$ km altitude with a time resolution of 1 h and an altitude resolution of 2 km when derived using the retrieval methods described in Stober et al. (2022). Atmospheric tides are derived by applying an adaptive spectral filter (ASF) (Baumgarten and Stober, 2019; Stober et al., 2020). The ASF is designed

to determine different tidal modes using rather short windows covering only 1-2 oscillations for each tidal mode, which makes the method ideal for such campaign-based datasets. The performance and applicability of the ASF were already successfully demonstrated by leveraging observations from EISCAT and the Nordic Meteor Radar Cluster (Günzkofer et al., 2022).

At high latitudes, the vertical propagation of diurnal tides is inhibited. However, upward-propagating semidiurnal tides gain large amplitudes and are the dominant tidal mode up to about 120 km altitude during most months of the year (Andrews et al.,

1987; Nozawa et al., 2010; Stober et al., 2021b; Günzkofer et al., 2022). Atmospheric tides are commonly considered to have an important impact on the neutral density in the lower thermosphere (Lieberman and Hays, 1994; Qian and Solomon, 2012; Truskowski et al., 2014; Maute et al., 2022; Yue et al., 2023). However, tides are most commonly measured as neutral wind or temperature oscillations with meteor radars and lidars. Figure 10 shows the amplitude of the semidiurnal atmospheric tide in the zonal and meridional neutral winds as measured with the Tromsø meteor radar.

The semidiurnal tide reached amplitudes of up to $\sim 40 \text{ ms}^{-1}$ during December 2022 in both zonal and meridional wind. This is close to the values given by the Global Scale Wave Model climatology (Hagan and Forbes, 2003). However, during the EISCAT Geminids campaign, the tidal amplitude was notably lower and reached amplitudes of only $\sim 20 \text{ ms}^{-1}$ and lower. Since the classical tidal theory usually applies logarithmic pressure units, neutral density oscillations are not directly described by the tidal equations (Andrews et al., 1987). However, measurements indicated that typical values for collision frequency

and neutral density oscillations due to tidal forcing at mid-latitudes are in the order of a factor two (Waldteufel, 1970; Monro et al., 1976). The neutral density variations observed in Figures 4 b) and 6 b) are in a range of factor two to five at 95 - 110 km altitude. Hence, at the lower measurement altitudes, the observed neutral density variations are comparable to the possible variations from tidal forcing. However, considering that semidiurnal tidal amplitudes decrease at higher latitudes and that the tidal amplitudes were generally decreased during the measurement campaign, it seems unlikely that the tidal forcing caused

neutral density variations in the order of a factor of two. This suggests that the relative importance of atmospheric tides for the ion-neutral collision frequency profile is small.

## 5.7   Joule heating impact

Assuming our interpretation that the observed changes of the ion-neutral collision frequency profile are the result of a neutral uplift due to ionospheric heating, other heating mechanisms need to be discussed. One of the most important heating mecha-

350 nisms in the lower thermosphere is Joule heating due to Pedersen currents which has been shown to cause significant neutral uplift in the lower thermosphere (Deng et al., 2011). The maximum Joule heating occurs at the Pedersen conductivity maximum at approximately 120 km altitude. The Joule heating drops rapidly at lower altitudes though there might still be a considerable impact at $100 - 110$ km altitude. For the December 2022 EISCAT measurements, the geomagnetic activity was consistently low with $Kp \leq 2$. Recent investigations showed that for such low geomagnetic activity, the Joule heating at 110 km altitude

reaches values in the order of $0.01\ \mu\text{W m}^{-2}$ (Baloukidis et al., 2023; Günzkofer et al., 2024) which is considerably lower than the estimated particle precipitation heating rates. However, the local geomagnetic activity over Tromsø can be higher than the global $Kp$ index suggests, as can be seen from the SME index in Figure 1. The local K index over Tromsø (Frøystein and Johnsen, 2024) reached values up to $K = 4$.

For May 2024, where the maximum geomagnetic activity reaches $Kp = 6$, the Joule heating at 110 km altitude can reach

$0.1\ \mu\text{W m}^{-2}$ or even higher values (Baloukidis et al., 2023; Günzkofer et al., 2024). Additionally, the increased ionization due to particle precipitation increases the Pedersen conductivity and consequently the Joule heating. Joule heating contributes to the upwelling of the neutral atmosphere above about 120 km (Deng et al., 2011). At 100 km altitude, Joule heating might contribute to ionospheric heating, especially for high geomagnetic activity. However, the Pedersen conductivity reduces rapidly below the altitude of its maximum. Therefore, particle precipitation is supposed to be the stronger heating mechanism at these altitudes

according to common literature. However, for strong precipitation conditions like during the investigated measurements, the common $Kp$ index estimate of Joule heating might be insufficient. Since the exact cause of the neutral density increase cannot be determined, a potential impact of Joule heating should be considered.

## 5.8 Rigid-sphere, Maxwell, and resonant collisions

The neutral particle density differences $\Delta n_n$ shown in Figures 4 b) and 6 b) are calculated from the $\nu_{in}$ profiles by applying Equation 1. It is assumed that ion-neutral collisions can be described as rigid-sphere collisions (Chapman, 1956). Another collision model often applied assumes Maxwell collisions of the ions and polarized neutrals (Schunk and Walker, 1971). However, it has been shown that the calculated neutral density profile is not impacted by the choice of the collision model for these altitudes (Günzkofer et al., 2023b). Therefore, applying the more simple rigid-sphere model is justified here. The plasma density $n_i$ in Equation 1 is commonly neglected for the calculations. This is reasonable at all altitudes, since even for the highest investigated altitudes at 110 km, the neutral density is larger than the plasma density by at least a factor of $10^6$.

Both approaches mentioned so far, rigid-sphere and Maxwell collisions, are so-called non-resonant collisions. However, resonance effects have to be considered for collisions between neutral particles and their first positive ions (in the MLT region, mainly $O_2$ and $O_2+$). Following Ieda (2020), $O_2$ and $O_2^+$ dominate predominantly resonant at ion temperatures $T_i > 600$ K (assuming ion and neutral temperatures to be the same in the MLT region). For the May 2024 measurements, $T_i > 600$ K was not reached at altitudes up to 110 km according to the UHF measurements (for which $T_i$ is fitted above 100 km). However, the December 2022 UHF measurements showed $T_i > 600$ K at 110 km for about 5% of all measurements. It should be noted that these $T_i > 600$ K measurements were entirely obtained during times of very low electron density and are hence subject to the remaining UHF data noise shown in Figure 7 b). Also, as already noted by (Ieda, 2020), since $O_2$ only makes up about 20% of the neutral particles in the MLT region, the impact on the total collision frequency is limited. Nonetheless, we repeated the analysis in Section 4 and included resonant collisions. The neutral density difference $\Delta n_n$ profiles shown in Figure 4 b) were unaffected by this recalculation since no $T_i > 600$ K conditions fell into the $N_{e,95} > 2 \cdot 10^{10}$ m$^{-3}$ domain. Hence, we conclude that the obtained $T_i > 600$ K are the result of low SNR ratios and, as already stated by Ieda (2020), resonant collisions can be neglected at altitudes up to 110 km.

## 5.9 The difference spectrum method

The difference spectrum method is described in Grassmann (1993b) and the limits of its application have been discussed in Günzkofer et al. (2023b). The main uncertainty of the difference spectrum method is related to the required $\beta$ parameter that is applied to compensate for technical differences between the UHF and VHF ISR when combining the two spectra. The $\beta$ parameter should be determined at F region altitudes where the ionosphere can be assumed to be collision-less. However, due to the different beam shapes of the UHF and VHF radars, the $\beta$ parameter varies, in fact, slightly with altitude (Günzkofer et al., 2023b). The *manda* pulse code applied for both EISCAT campaigns analyzed in this paper allows measurements up to 200 km altitude, which is below the F-layer and, thus, the $beta$ parameter can only be derived with some margin of uncertainty. In this study, the $\beta$ parameter is determined at this maximum altitude where the ratio of ion-neutral collision to gyro-frequency is approximately 0.02 taking into account the climatological average. This is a drawback of the *manda* pulse code compared to the *beata* mode which covers higher altitudes above 200 km and, hence the assumption of a collision-less plasma is better satisfied at these higher altitudes making the analysis more resilient for this EISCAT mode.

## 5.10 Quantification of particle precipitation

To investigate the impact of particle precipitation on the $\nu_{in}$ profile, we binned the obtained collision frequency measurements with the electron density at 95 km altitude $N_{e,95}$. Applying $N_{e,95}$ as quantification for the particle precipitation impact assumes that it is the dominant ionization mechanism at this altitude. Following Fang et al. (2010, 2013), the particle precipitation impact at 95 km altitude is mainly carried by electrons with energies of about $10 - 100$ keV and protons with energies of about 1 MeV. For particle precipitation energy rates of 1 mW m$^{-2}$, the ionization rates due to particle precipitation are in the order of $10^{10}$ m$^{-3}$ s$^{-1}$. Especially during December, when the solar elevation angle is very low at high northern latitudes, this should exceed the photoionization rate (Baumjohann and Treumann, 1996). The solar elevation angle is considerably higher in May at the Tromsø geographic latitude. Therefore, photoionization rates during daytime measurements of the May 2024 EISCAT campaign cannot be neglected and the quantification of particle precipitation impact by $N_{e,95}$ introduces additional uncertainties.

Our analysis is based on steady-state conditions during the ISR experiment dwell time of 120 s, which is presumably only partly justified as the energy transfer happens on much shorter time scales for individual collisions. However, this seems to be justified at least statistically for the ensemble of all precipitating particles within the observation volume. Most ionospheric dynamics take place on larger time scales but it has been shown for frictional heating processes that shorter scales do contribute as well (Brekke and Kamide, 1996). A shorter dwell time for the radar experiments is not feasible for observing E-region dynamics due to the lower total electron densities at the transition region between D- and E-region.

In Figure 2 a), it can be seen that occasionally the electron density below 100 km appears to be higher than at higher altitudes. In these cases, the $N_{e,95} > 2 \cdot 10^{10}$ m$^{-3}$ ion-collision frequency profiles may underlie the data noise issues shown in Figure 7 that cause the enormous uncertainty of the $N_{e,95} < 1 \cdot 10^{10}$ m$^{-3}$ profile in Figure 4 a). However, only for about 0.5% of the total measurements, it was found that $N_{e,95} > 2 \cdot 10^{10}$ m$^{-3}$ while the electron density at 105 km $N_{e,105} < 1 \cdot 10^{10}$ m$^{-3}$. We repeated the analysis in Section 4 with excluding all measurements with $N_{e,95} > N_{e,105}$ but obtained nearly identical results.

## 6 Conclusions

We studied the variation of the ion-neutral collision frequency, measured by dual-frequency ISR observations during two measurement campaigns, one during the Geminid meteor shower in December 2022 and another during a solar energetic particle event in May 2024. We found a distinct diurnal variation of the $\nu_{in}$ profile which indicates a significant deviation from the climatology depending on the time of the day. Applying the electron density at 95 km altitude $N_{e,95}$ as a quantification for the strength of particle precipitation, we showed that there is a connection of the ion-neutral collision frequency at 90 - 110 km altitude and particle precipitation. Below about 100 km altitude, the ion-neutral collision frequency decreases for large $N_{e,95}$ while above about 100 km altitude, $\nu_{in}$ is increased. However, the collision frequency profiles measured under low electron density conditions showed large uncertainties and therefore only the measurements for high ionization can be assumed reliable.

Assuming a rigid-sphere ion-neutral collision model, the difference of the neutral particle density profiles $n_n$ for $N_{e,95} >$

$2 \cdot 10^{10}$ m$^{-3}$ from the climatology was calculated. Our interpretation of the $\nu_{in}$ profile variation for changing $N_{e,95}$ is that the heating due to the precipitating particles causes an up-welling of the neutral atmosphere. This is also observed in a second ISR measurement campaign that was specifically conducted during a SEP event and therefore exhibits continuously high $N_{e,95}$ values. We found that during the SEP event, the ion-neutral collision profiles resembled the profiles measured for the highest $N_{e,95}$ values of the Geminid campaign. The neutral particle density profiles measured during the SEP event exhibit a similar decrease/increase of neutral particle density though the uplift is shifted to lower altitudes by about 2 -3 km. We interpret this as the result of a higher energy of the precipitating particles. Changes in the ion-neutral collision frequency profile caused by up-welling due to ionospheric heating have been previously reported and discussed (Nygrén, 1996; Oyama et al., 2012). However, we discussed alternative explanations for the observed changes of the ion-neutral collision frequency like Joule heating and atmospheric tides as well as data quality issues due to the generally low electron density in the MLT region. Neutral winds along tilted isobars have been shown to cause vertical winds of a similar velocity as the estimated neutral uplift due to particle precipitation heating (Oyama et al., 2008). The observed collision frequency profiles cannot be conclusively linked to a neutral uplift caused by particle precipitation heating though a correlation has been shown in this paper.

We estimated the physical impact of the inferred ion-neutral collision frequency profiles and found a major impact of the $\nu_{in}$ uncertainties on the energy balance of the observed atmospheric uplift. This indicates that exact quantitative conclusions have to be drawn carefully since both physically possible and impossible (in terms of energy balance) atmospheric changes are within the measurement uncertainty range. Furthermore, we performed a sensitivity analysis highlighting how different climatology profiles used to initialize the dual-frequency fitting can impact the $\nu_{in}$ profiles for low SNR measurements often related to low electron densities. This revealed that the $\nu_{in}$ profiles obtained with the difference spectrum method are considerably impacted by the *a priori* collision profile below 95 km altitude. This also needs to be considered when drawing conclusion about the observed neutral uplift.

This study has shown how neutral atmosphere dynamics in the ionospheric dynamo region can be investigated by ion-neutral collision frequency measurements with dual-frequency ISRs. Though such measurements are rare so far, they provide a promising method to investigate the impact of ionospheric processes on the neutral atmosphere.

Future dual-frequency ISR campaigns should aim to observe different storm conditions in addition to SEP events, e.g. following coronal mass ejections and unusually strong substorms and superstorms. Similarly, the impact of other ionospheric heating mechanisms, especially Joule heating, on the neutral atmosphere can be studied. Dual-frequency ISR campaigns following strong geomagnetic storms, as they can be expected during the current solar maximum, could show a similar up-welling of the neutral atmosphere as caused by particle precipitation. Since the neutral atmosphere above 100 km altitude is generally difficult to measure, dual-frequency ISR measurements might also give further insight into seasonal changes, such as those caused by variations of tidal and gravity wave activity around the spring and fall equinoxes. Lastly, the difference spectrum method has been applied to multiple dual-frequency ISR campaigns so far and appears to be a reliable analysis method for these experiments. However, a definite verification of the method is not possible with dual-frequency measurements. An all-time unique opportunity for triple-frequency ISR measurements might be possible when the EISCAT UHF and VHF ISRs are operated together with the upcoming EISCAT_3D radar (McCrea et al., 2015). The EISCAT_3D radar will be operated at

nearly the same radar frequency as the EISCAT VHF ISR but the beam shape will resemble that of the UHF ISR, which permits
the quantification of the suspected impact of the different beam shapes and the corresponding differences in the observation
volumes between the VHF and UHF ISRs.

*Data availability.* The data are available under the Creative Commons Attribution 4.0 International license at https://doi.org/10.5281/zenodo.
14646603 (Günzkofer et al., 2025) Version v2.

*Author contributions.* FG performed the data analysis and wrote large parts of the paper. GS and CB contributed to the interpretation of the
analysis and GS wrote parts of the paper. JK and DRT were PIs of the analyzed EISCAT campaigns. AT contributed to the initial single-
frequency analysis of the EISCAT measurements. NG and MT are PIs of the Tromsø meteor radar. All authors provided feedback and were
involved in revising the manuscript.

*Competing interests.* GS and CB are handling editors at Annales Geophysicae.

*Acknowledgements.* EISCAT is an international association supported by research organizations in China (CRIRP), Finland (SA), Japan
(NIPR and ISEE), Norway (NFR), Sweden (VR), and the United Kingdom (UKRI). GS is a member of the Oeschger Center for Cli-
mate Change Research (OCCR). DRT is supported by the United Kingdom Natural Environment Research Council (NERC) DRIIVE
(NE/W003317/1) grant. May 2024 EISCAT observations in this study were conducted as an allocation of UK NERC EISCAT operations.
FG acknowledges helpful discussions during the 21st International EISCAT Symposium 2024. We gratefully acknowledge the SuperMAG
collaborators (https://supermag.jhuapl.edu/info/?page=acknowledgement).

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
