# Peer review of "Indications for particle precipitation impact on the ion-neutral collision frequency analyzed with EISCAT measurements"

_EGUsphere, 2024_

## Author Response (AR1)

**Reviewer 1**

**Comment 1**

Reviewer:

This study analyzed two datasets from collaborative measurements with the EISCAT UHF and VHF radars in Tromso, Norway, during moderately active conditions characterized by hard particle precipitation. By applying the difference spectrum fitting method developed in a previous study, the height profiles of the ion-neutral collision frequency in the MLT region were derived. The ion-neutral collision frequency is a critical physical parameter for understanding the ionosphere-thermosphere coupling process; however, its features have not been fully elucidated. Based on the derivation, this study presented the impacts of precipitating particle forcing on the ion-neutral collision frequency and discussed plausible causalities that produce variations in the collision frequency. However, the explanation of the calculation results and physical mechanism requires more rigorous and comprehensive analysis of data and consideration from multiple perspectives, as the current text is biased toward only certain aspects or interprets phenomena in an overly simplistic manner. These issues are summarized in the major comments. The accuracy of the calculations and discussion is significantly low, and the results of this study do not meet the standards for publication. Since the quality is unlikely to be improved by reanalysis, as mentioned in Major comment 1, the recommendation for the editor is to reject.

Authors:

We thank the referee for taking the time to read and evaluate our paper.

[Major comments]

Reviewer:

1. Artificial discontinuity at 100 km altitude

In examining Figure 2b, a discontinuity in the collision frequency is observed at approximately 100 km altitude. This discrepancy is more evident in the line plot presented in Figure 3. The quick-look figure of the EISCAT measurements available in the Madrigal database clearly indicates a significant difference in the noise level of the UHF-measured electron density above and below 100 km. One of the conclusions of this study addresses the differential effects of thermospheric density variations at altitudes above 100 km. Based on the results presented in Figures 4b and 6b, this study corroborates that thermospheric density decreases and increases at altitudes below and above 100 km, respectively. However, the validity of this conclusion is subject to scrutiny given the apparent disparity in the quality of the EISCAT measurements, which constitute the primary data used to calculate the collision frequency.

While the manda pulse code employed in radar measurements exhibits an advantage in measuring lower-altitude electron density with high range resolution, it demonstrates reduced accuracy in measuring ionospheric parameters in the E region and above compared to other pulse codes. The analysis presented in this study does not account for this characteristic (at least, no explanation is provided in the text).

Authors:

The mentioned data noise issue is very important and we address the raised concerns by additional analysis involving EISCAT employees. First, we changed the analysis settings to 120s integration windows and the altitude gates to 60 logarithmically-spaced gates from 50 – 200 km altitude. In the region of interest, this results in an altitude resolution of about 3 km. We devoted a new Setion 5.1 to the noise issue in the revised manuscript, where the mentioned discontinuity at 100 km is shown and discussed (it is mainly caused by a change in the a priori settings of UHF manda analysis). It can be seen that for the adjusted altitude gates and integration time, the strong data noise above 100 km is only present for the very lowest electron densities.

Reviewer:

As illustrated in Figure 3, the collision frequency below 95-100 km altitude exhibits a strong dependence on a priori. The authors assert that the altitude region where the influence of the a priori is significant extends up to 95 km, and they have hatched this region with gray shadows (Figures 4 and 6). Figure 3 demonstrates that the upper altitude limit is contingent upon the choice of a priori, and the a priori should be considered dominant up to 97-100 km altitude. The noise level of the EISCAT measurements is elevated for collision frequencies above 100 km altitude, as noted above. Given these considerations, the calculated results for any altitude range are deemed unreliable for this dataset, and there appears to be no justification for the discussion and conclusions based on these ambiguous results. Consequently, the results, arguments, and conclusions drawn in this study based on the observed data are considered unreliable. It is recommended to utilize EISCAT UHF and VHF simultaneous observation data employing another pulse code rather than manda.

Authors:

After mitigating the noise issue by re-analyzing the data with an increased integration time and increased range gates, the results are substantiated and are less dependent on a priori information. However, we would also like to state that the high electron density profiles shown in the original manuscript were reliable, as it can be seen in Figure 7 of the revised manuscript that the data noise was low during times of strong ionization (strong particle precipitation). Given that the new analysis mitigated the noise issue for the altitude region of concern, we disagree to the reviewer conclusions.

Reviewer:

2. Physical parameters to affect on the ion-neutral collision frequency

The collision frequency is proportional to the thermospheric and ionospheric-plasma densities, as demonstrated in Equation 1. However, this equation represents a simplified model and is also dependent on temperature (Prolss, Physics of the Earth's Space Environment, 2004). Given that the event was observed during periods of high geomagnetic activity, it is logical that the temperature of the thermosphere and ionosphere increased due to particle heating and Joule heating. Rather than restricting the analysis to attributing all collision-frequency variations to the density variations, it would have been more comprehensive to incorporate temperature effects in this study.

The variation in thermospheric density is estimated from the increase or decrease in collision frequency (Figure 4b). As demonstrated in Equation 1, the collision frequency is a function of thermospheric density; however, it is also dependent on ionospheric plasma density. Lines 167-170 in the text briefly describe the derivation method, yet it remains unclear whether the thermospheric density was calculated considering the electron density measured by the EISCAT radar. If this factor had not been considered, it should have been incorporated into the calculations. If the collision

frequency has been calculated taking this into account, the error in the atmospheric density should be determined by considering the error in the measured electron density and discussing its significance in relation to the magnitude of the thermospheric density variations.

Authors:

A discussion of collision frequencies has been added to Section 5.7 of the revised manuscript: The reviewer is correct in pointing out that other parameters than the neutral density can potentially impact the collision frequency. However, as described by Ieda, A. (2020) "Ion-neutral collision frequencies for calculating ionospheric conductivity", only the resonant ion-neutral collisions are temperature dependent. Non-resonant collisions are not temperature-dependent and dominate below 600 K. However, we agree that the T<600 K condition does not trivially hold during intensified particle precipitation. We thank the referee for pointing this out to us. We found T>600 K below 110 km altitude during about 5% of the December 2022 measurements. However, these were the measurements with the lowest electron density and consequently highest uncertainties. Therefore, it is arguable that the temperature stayed below T<600 K during out measurements below 110 km (as also stated by Ieda, 2020). Furthermore, including resonant collisions would not affect the high precipitation collision profiles (the low precipitation profile is actually also not affected significantly). Hence, neglecting resonant collisions is a reasonable assumption below 110 km (as concluded by Ieda, 2020).

The plasma density impact on the ion-neutral collision frequency is apparent from Equation 1. Considering the relativ number densities between a thermalized plasma and the neutrals, the collision frequency is dominated by the neutral part as ions form only a minor contribution to the overall number density. However, we agree to the reviewer that this is only true for a thermalized plasma. As soon as there is a significant heating of the plasma, the collision frequency is affected (see Stober et al., 2023). A brief discussion of this has been added to the manuscript.

In general, ion-neutral collisions can be described in multiple ways, e.g. as rigid-sphere collisions Chapman (1956) "The Electrical Conductivity of the Ionosphere: a Review." which results in Equation 1, or as non-resonant Maxwell collisions Schunk and Walker (1971) "Transport Processes in the E region of the Ionosphere". As shown in our previous publication on the difference spectrum method (Günzkofer et al., 2023), these two approaches result in very similar neutral density profiles. Thomas et al. (2024) "Dregion ion-neutral collision frequency observed by incoherent scatter spectral width combined with LIDAR measurements" in turn showed that the non-resonant collision frequency according to Ieda (2020) is nearly equivalent to the Schunk and Walker (1971) equation. As stated above, other parameters become significant for resonant ion-neutral collision frequencies (dominant at T>600 K).

Reviewer:

3. Feature of the vertical motion in the lower thermosphere

It was previously mentioned that variations in collision frequency were not exclusively attributable to fluctuations in the thermospheric density. Even if density variation is presumed to be the primary factor governing the collision frequency variation, the explanation provided in the text would not align with the characteristics of vertical motion in the lower thermosphere. When examining the vertical motion of the lower thermosphere, it is essential to consider horizontal motion, particularly along isobars. In the lower thermosphere, where horizontal motion predominates, an upward displacement of the isobar in the heated region induces an apparent vertical component of the wind in geographic coordinates due to thermospheric winds flowing along the isobar. In instances of

intense localized heating over brief periods, upwelling across the isobar may occur, but this phenomenon dissipates rapidly in conjunction with vertical oscillations. Upon cessation of heating, the isobar expansion terminates, and the apparent vertical component diminishes. However, during a transition process under force balance between buoyancy and gravity, the atmosphere undergoes oscillation (i.e., atmospheric gravity waves are generated), and vertical motion may persist for a duration. Nevertheless, this phenomenon does not result in an increase in the spatiotemporal mean density.

Figures 4b and 6b indicate that the atmospheric density increased above 100 km altitude irrespective of the electron density level at 95 km altitude employed in this study. During periods of high geomagnetic activity, characterized by high electron density, the atmospheric density above 100 km altitude may increase. However, the intermittent increase in electron density over the two and a half days of December 13-15 suggests that it is improbable that the energy flow from the magnetosphere to the polar thermosphere/ionosphere is sustained at a sufficient level to support the density increase. Considering that particle heating should have occurred at the same location as the aurora, and given the likely structured nature of the aurora, it is implausible to assume constant upwelling in the EISCAT radar beam, although Major comments 1 and 2 elucidate the unreliability of the calculation results. Even if the calculation results capture some degree of nature, the physical interpretation of this study cannot be objectively substantiated.

Authors:

The referee states that localized heating can cause upwelling across the isobars (resulting in an increase of thermosphere density) which diminishes shortly after the heating ceases. Therefore, the heating does not cause a general, persistent increase of the thermospheric density. We agree with the referee and see this as the main conclusion of our paper.

 "Figures 4b and 6b indicate that the atmospheric density increased above 100 km altitude irrespective of the electron density level at 95 km altitude employed in this study."

We would argue that the opposite is the case. Figures 4b and 6b show that the atmospheric density above 100 km altitude is increased for $N_{e,95} > 2 \cdot 10^{10}$ ($1 \cdot 10^{11}$) $m^{-3}$ compared to $N_{e,95} < 10^{10}\ m^{-3}$.

"During periods of high geomagnetic activity, characterized by high electron density, the atmospheric density above 100 km altitude may increase. However, the intermittent increase in electron density over the two and a half days of December 13-15 suggests that it is improbable that the energy flow from the magnetosphere to the polar thermosphere/ionosphere is sustained at a sufficient level to support the density increase."

As argued in the manuscript, the majority of $N_{e,95} > 2 \cdot 10^{10}\ m^{-3}$ conditions during December 13-15 2022 occur during the night from Dec 14 to 15 where the particle precipitation heating sustains over a longer time interval. As shown in Figure 4 a, the increase of atmospheric density is restricted to these conditions and significantly lower densities are found for non-heating conditions. This is equivalent to the referee's statement that the upwelling of the atmosphere ceases quickly after the heating stops. "…, it is implausible to assume constant upwelling in the EISCAT radar beam"

We agree with this statement and do not see how our results would indicate a constant upwelling of the atmosphere. Our results exhibit that the upwelling is only found for strong heating conditions, quantified by the electron density at 95 km altitude. The duration of the auroral precipitation events presented in (Grandin et al. (2024) "Statistical comparison of electron precipitation during auroral breakups occurring either near the open-closed field line boundary or in the central part of the

auroral oval") of roughly 20 min is slightly below the required atmosphere reaction time discussed in Section 5.4 of our paper. However, Kurihara et al., 2009 and Deng et al., 2011 showed that usually the actual reaction time is considerably faster than calculated by the simple estimate in Section 5.4.

[Minor comments]

Reviewer:

L3: "momentum transport" should be revised to "momentum transfer."

Authors:

Done as suggested.

Reviewer:

L13: "different a priori collision frequency profiles" may be better to say "various a priori collision frequency profiles."

Authors:

Done as suggested.

Reviewer:

L48-49: A magnetometer in Tromso, which can be checked at the IMAGE webpage, presents substorms during the high electron density periods selected in this study. The effects of Joule heating should be discussed, along with those of particle heating.

Authors:

A discussion of the Joule heating impact following (Deng et al., 2011; Baloukidis et al., 2023 and Günzkofer et al., 2024) has been added in Section 5.6 of the revised manuscript.

Reviewer:

L60-61: According to the EISCAT QL from the Madrigal database, the transmitter powers of the UHF and VHF radars have not reached these numbers.

Authors:

It was clarified that these are the maximum possible transmission powers which were not reached during these campaigns.

Reviewer:

L66: "06-15 UT is analyzed" should be revised as "06-15 UT, is analyzed."

Authors:

Done as suggested.

Reviewer:

Section 5.1: The effects of the ambiguity of the beata parameter on the derived collision frequency should be evaluated in a quantitative manner.

Authors:

The effect of the beta parameter on the ion-neutral collision is not linear and is difficult to describe in a simple comprehensive statement. A comparison of collision frequency profiles for different beta parameters is shown in (Günzkofer et al., "Difference spectrum fitting of the ion-neutral collision frequency from dual-frequency EISCAT measurements", 2023). A quantitative evaluation of the beta dependency is beyond the scope of this paper and was already presented previously. A full comprehensive evaluation would require triple frequency observations with identical beam pattern. This is not available for EISCAT.

Reviewer:

L222-224: The explanation is incorrect, according to Figure 3.

Author:

We corrected the explanation (correct: the jump is found when the EISCAT a priori profile is used to initialize the fit and when the NRLMSIS profile is used but then only for noon and dusk sectors). Due to the new analysis parameters, there are no more jumps in the profiles.

Reviewer:

Section 5.5: The meteor radar measures the horizontal wind, but what this study mentions is the density. To apply the meteor radar measurements, experimental evidence or theoretical support is required to link the wind and density in advance.

Author:

The connection between tidal wind and density amplitudes was insufficient in the previous text. We added a more extensive discussion together with references to common literature on this topic (Liebermann and Hays, 1994; Qian and Solomon, 2012; Truskowski et al., 2014; Maute et al., 2022; Yue et al., 2023). Generally, a direct link is difficult since the classical tidal theory applies pressure coordinates and therefore does not explicitly include neutral densities. However, there are some past measurements of collision frequency/neutral density connected to tidal forcing. These were applied to estimate the expected neutral density variation from tidal forcing which is presumably lower than the variation observed in Figures 4 b) and 6b).

**Comment 2**

The reviewer anticipated quantitative counterarguments rather than a vague qualitative response; however, it appears that more concrete evidence should have been provided in the initial comment. A brief analysis of the same EISCAT measurements examined in this study has been conducted, and these findings will be referenced to elucidate the reviewer's perspective.

Reviewer:

The attached Figure C1 was generated utilizing UHF and VHF radar data for December 13-15, 2022, obtained from the Madrigal database. The observed discontinuity is present exclusively in the UHF radar data (Figure C1a). This phenomenon is a recurrent issue that may arise when the manda pulse code is applied to the UHF radar during periods of low electron density, and the discontinuity is an

artifact of this technical limitation. It is evident that the discontinuity observed in the collision frequency, as illustrated in Figure

2b of this study, originates from the UHF measurements. As the UHF radar measurements are not presented in this study, it is challenging for readers to discern the underlying cause. The figure should have been included to elucidate the reason clearly, thereby preventing any potential suspicion of intentional concealment.

Authors:

See responses above and Section 5.1 of the revised manuscript.

Reviewer:

In this study, the electron density at an altitude of 95 km was utilized as a parameter to categorize the calculated collision frequencies. Specifically, the electron density at an altitude of 95 km was employed to ascertain the presence of a discontinuity effect at an altitude of 100 km. Consequently, it is imperative that the validity of this methodology be evaluated quantitatively and rigorously. The appropriateness of the threshold value ($2 \times 10^{10}$ m-3) should have been quantitatively assessed through a comparative analysis of results derived using alternative thresholds.

Authors:

We created a histogram of $N_{e,95}$ (see below). Applying logarithmic bin edges ($10^x$ with x=9.0, 9.1, 9.,2, …, 11.0), a bimodal distribution is noticeable. The threshold for low electron densities is slightly larger than the left maximum thereby enclosing the majority of time points. The threshold for high electron density is approximately at the minimum in between the maxima, thereby ensuring the stability of the obtained profiles for small variations of the threshold. This can be seen in the plot below of four collision frequency profiles obtained for different thresholds of high $N_{e,95}$. It can be seen that the variation of collision frequency becomes more pronounced the higher the threshold. The original choice of thresholds was done qualitatively but we hope that this quantitative analysis provides the requested assessment of appropriateness.

[Figure]

[Figure]

Reviewer:

Figures C1b and C1d illustrate the EISCAT-measured electron density for periods when the electron density at an altitude of 95 km exceeds 2x10^10 m-3. For the majority of the selected periods, the electron density in this altitude range (85-110 km) exhibits an increase with altitude; however, this trend is not consistent across all time points (e.g., approximately 12 UT on 2022.Dec.13, 18 UT on 2022.Dec.14, 9-12 UT on 2022.Dec.14). This observation suggests that the measurement uncertainty in the upper layers may be greater than that in the lower layers, and adequate measurement accuracy at an altitude of 95 km may not necessarily ensure the accuracy of measurements in the upper layers. It is imperative to evaluate the validity of classification based on the 95 km altitude electron density for these exceptional cases.

Authors:

This is an important observation for which we are thankful. We found that $N_{e,105} < N_{e,95}$ is the case for about 25% of the measurements during December 2022. However, only in 1% of the measurements we found that $N_{e,95} > 2 \cdot 10^{10} \, m^{-3}$ while $N_{e,105} < 2 \cdot 10^{10} \, m^{-3}$ which would impact the reliability of the high electron density collision frequency measurements. For safety, we redid the analysis and excluded all measurements with $N_{e,105} < N_{e,95}$ and obtained the Figure below which is nearly identical to Figure 4 a) (the collision frequencies are a bit larger for low and medium ionization but remain mostly unreliable). We also added a discussion of this in Section 5.9 of the revised manuscript

[Figure]

Reviewer:

Assuming the aforementioned threshold and data selection method are appropriate, the time intervals for which reliable collision frequencies can be calculated are limited to those depicted in Figures C1b and C1d. Consequently, the collision frequencies for other time intervals lack reliability. Figure 2b presents collision frequencies for all time periods, potentially leading readers to erroneously conclude that collision frequency can be derived even for low electron densities. Figure 3 is categorized by time of day; however, it does not account for data classification based on the method applied in this study and calculates the average collision frequencies without considering the confidence level of the EISCAT data.

Authors:

This is a fair point. We added the uncertainty range for the low electron density collision frequency profile in Figure 4 a) to illustrate that these profiles are not very reliable. We also added a clear statement that reliable ion-neutral collision frequency measurements are only possible for sufficient ionization. Therefore, the climatology profiles were applied as baseline for the neutral density difference profiles in Figure 4 b) and 6b).

Reviewer:

Additional issues requiring quantitative analysis include heating due to particle precipitation and upwelling of the thermosphere. The text at L126-128 states that "Presumably, the atmospheric heating due to precipitating particles causes uplift of the neutral atmosphere, which in turn results in an increased neutral particle density and consequently an enhanced ion-neutral collision frequency for these altitudes." This statement implies that the particle heating energy generated during a particle precipitation event invariably produces sufficient thermal energy to cause upward flow in the lower thermosphere. The use of "Presumably" is inadequate to justify this assertion. This statement should be omitted. The conclusions that can be reliably drawn from the results of this study are as follows: 1) the collision frequency can be calculated by this method only during events of particle precipitation and high ionospheric density, and 2) the calculated collision frequency during this limited time interval is lower at lower layers and higher at upper layers compared to the climatological results. Regarding point (2), this study discusses thermospheric density fluctuations; however, these may be apparent fluctuations resulting from uncertainties in the climatological results. Even if density fluctuations do occur, they are not necessarily caused by upwelling generated by particle heating; they may be attributed to Joule heating, or they may be the result of advection from other regions. It is questionable whether the discussion in this study encompasses all possibilities. The conclusion that "the particle precipitation heating causes a significant uplift of neutral gas between about 90-110 km altitude" appears to be speculative and is not a comprehensive assessment based on quantitative consideration of the results.

Authors:

We agree that our original manuscript did not sufficiently reflect the uncertainties from our results and interpretation. We added a discussion of the Joule heating impact in Section 5.6 and also carefully revised our conclusions to reflect the reviewer's largely correct criticism.

Reviewer:

Based on the aforementioned analysis, this study exhibits critical deficiencies in the following areas: rigorous analysis considering data quality, methodology for assessing the confidence level of the calculated collision frequencies, and comprehensive discussion of the results and conclusions. Consequently, it is recommended that the editor rejects the study, as previously outlined in the initial comments. It should be noted that this assessment does not invalidate the method employed to derive the collision frequency.

Authors:

We thank the reviewer for explicitly stating that the method to derive collision frequencies is principally valid. We hope that with the reduced data noise issue due to the adjusted analysis settings and a more careful discussion of the obtained results, the paper is now suitable. Due to the extremely low number of studies addressing the collision frequency in this region and therefore the absence of a ground truth, we see our paper as a relevant contribution to the field.

**Reviewer 2**

Reviewer:

The authors use simultaneous measurements from the EISCAT UHF and VHF incoherent scatter radars to examine the dependence of the ion-neutral collision frequency on particle precipitation. They purport to show a significant impact and that the particle heating causes a significant uplift of neutral gas between 90 and 110km.

This was a fascinating and worthy attempt to use the multifrequency data to extract information about the underlying neutral atmosphere. The technique and type of analysis involved was sensible and interesting. However, inspection of the underlying data has made me concerned as there seems to be some systematic noise issues that may persist even at high density levels and need to be properly identified and discussed. I feel the current discussion is not enough and misses some worrying features in the data (e.g. the UHF noise band from 100-110 km, the heightened noise spikes at all altitudes in the VHF).

I think that the fairest approach would be to ask the authors to conduct an analysis of the underlying data noise, speak to EISCAT staff about the measurements to check if there were additional issues (the system temperature on the VHF seemed rather high at times). This will give everyone more confidence in the data and the analysis. This could be done through rejection and request resubmission, though I am minded to recommend a major revision instead, unless the authors feel the timescale to carry this out would be too long? Even if there are fundamental issues with the underlying data there may be solutions for mitigating these that could make the study worth publishing even as those flaws are acknowledged.

Authors:

We thank the reviewer for taking the time to read our paper and their thorough evaluation.

Major comments

Reviewer:

Data noise:

The underlying data as presented in figure 2 is somewhat noisy. I worry this may impact severely the analysis that the authors have done. Did they consider increasing the integration time and/or size of the range gates? I note that the noise issue is most pronounced in the UHF data, inspecting the MADRIGAL summary plots for 13 -15 December 2022) the UHF shows a persistent band of increased noise from 100-110 km altitude, which is worrying. The VHF shows highly noisy data below 110 km.

Authors:

We contacted EISCAT staff and adjusted the settings of our GUISDAP analysis according to their recommendations, which were similar to what the reviewer suggests:

We increased the integration time from 60s to 120s and increased the size of the range gates to 60 logarithmically spaced bins from 50 – 200 km. This considerably decreased the noise as shown in Section 5.1 and Figure 7 of the revised manuscript. We added a discussion on the noise issue in section 5.1. This discussion is based on recommendations from EISCAT stuff. We also added EISCAT member to the list of authors.

Reviewer:

One would expect an increased level of relative noise in regions of lower density (i.e. low altitude) but for the data here it seems remarkably high with large multi-gate spikes. It is harder to tell if this persists when density is high though I note in figure 2b the first large precipitation event after noon on the 13/12/2022 there is a noticeable step change in the collision frequency that continues through the period. It appears to reduce during periods of very increased density, though this could be a color scale effect and so I am still wary.  That the discontinuity manifests in the collision frequency data from the difference spectrum fitting method as well as in the GUISDAP analysed data suggests that it is an underlying data issue and not simply an analysis problem. If this is the case it would impact all of the data and so restricting analysis just to the high-density regions would not completely mitigate the problem. This needs thorough investigation.

Authors:

The high noise issue is caused by the low electron densities in combination with the narrow range gates of the manda pulse code and, to the knowledge, there is no underlying issue with EISCAT manda measurements in general. We mitigated the issues by reprocessing the data using an increased width of the range gates. This removed/suppressed the spikes and discontinuities in the analyzed data of the region of interest.

Reviewer:

I wondered if this may be that this is simply indicative of the transition from the a priori value to the measured value. However, this does not resolve the issue of the band of noise in the UHF density data that is suggestive of a bigger problem (and in the VHF manifesting in a different way).

Author:

The transition of a priori settings impacts the onset of the high noise band at 100 km altitude. The upper edge of the band is caused by the change of altitude gate settings. With the new altitude gate setting, the data noise band is significantly reduced and only remains for times of very low electron density.

Reviewer:

The issue with the data is harder to see in May 2024, though the low altitude VHF data still shows unusual noise structure. I had a quick look at some older manda data in the Madrigal archive and the noise structure looks different (this is anecdotal and far from comprehensive), which makes me suspicious of these recent observations and whether there is an underlying issue in the latest version of the manda code. Did the authors discuss the data with the EISCAT staff? It would be worth identifying if this is a problem

with the current implementation of manda or something else (some sort of aliasing problem with more contaminated range gates at higher altitudes?)

Authors:

As far as we know, there is no specific issue with the manda code and its implementation. All noise issues are caused by the increased range resolution of the manda mode. We checked the data noise for the May campaign equivalent to Section 5.1 in the revised manuscript and attached the relative uncertainties of the VHF measurements with the standard analysis settings (see below). A discontinuity is found at 110 km but the relative uncertainties are overall very low above 90 km. Also, this discontinuity vanished with out adjusted analysis settings.

[Figure]

Reviewer:

I think it is essential for the authors to conduct an analysis of the uncertainties and the level of variability (noise) in the data to identify any trends, or sharp changes with altitude and underlying density level. Expanding the analysis upwards beyond 110 km might be instructive in terms of identifying any additional discontinuities that would point to a problem in the noise level.

Authors:

We added a detailed and comprehensive discussion of the noise issues in section 5.1.

Minor to moderate comments:

Reviewer:

Line 24: the assumption is made that the ion density is equal to the electron density; this is generally true at higher altitude but as one moves lower in the D region the negative ion density increases such that it may become appreciable (especially in darkness). How would this affect the use of Equation 1.

Authors:

Generally, it is assumed that negative ions can be neglected above 80 km altitude. However, even if considering the presence of negative ions, Eq. 1 can be applied as it is derived under the assumption of rigid-sphere collisions, i.e. no electrodynamic interaction of ions and neutral, and hence the sign of ion charge is of no importance to the collision process. Since the ion density in Eq. 1 is usually neglected, because it is much lower than the neutral density, negative ions are supposed to have a minimal impact on the collision frequency. A discussion of the validity of the rigid-sphere collision assumption is added in Section 5.7 of the revised manuscript.

Reviewer:

Line 69: "the UHF and VHF radars were operated in the manda zenith mode, also known as the Common Programme (CP) 6" – this is not accurate, it is similar to the CP6 mode. "Common Programme 6" has a very specific meaning beyond the pulse code and pointing direction. It would be more correct to say that "the UHF and VHF radars were operated pointed in the zenith using the manda pulse code, which is optimised for high resolution D-region measurements (as used in the EISCAT Common Programme 6)"

Authors:

Changed as suggested.

Reviewer:

Line 130: a little more information on how the EISCAT climatology is constructed would be welcome to the reader who may not be familiar with EISCAT and climatologies. Given the way in which EISCAT data presents more explanation on how it is a straight line would be welcome. I assume this also indicates that the standard EISCAT analysis is not using NRLMSIS to provide the a priori data to the fit otherwise the two lines should be the same. Apologies if I have got this muddled but I think that demonstrates the need for further information on how this is constructed.

Authors:

A brief explanation has been added to the paper: The climatology for collision frequencies in this case is taken from the CIRA2014 model. The climatology model depends on the GUISDAP version and installation. The GUISDAP analysis for this paper was conducted on a Windows system, which does not allow for a full GUISDAP installation, and therefore older climatology models are used (IRI2012 and CIRA2014). A full installation of the latest GUISDAP version, which is also used to obtain the madrigal data, would apply climatology models IRI2020 and NRLMSIS2.1.

Reviewer:

Line 164: The authors suggest that the detected uplift is due to direct particle heating. This may play a sizeable role; however, did the authors consider the role of Joule heating? This tends to maximise above the region of interest in this work but Baloukidis et al., 2023 (https://doi.org/10.1029/2023JA031526) showed that there can be appreciable heating towards the top of the region of interest and Deng et al., 2011 (https://doi.org/10.1029/2010JA016019) suggested that the uplift can start at lower altitudes. I think it unlikely that Joule heating alone could cause the potentially observed

upwelling at the reported altitudes, but it is worth discussing this possibility with reference to the potential mechanisms and qualified assessment.

Authors:

A discussion of the Joule heating rates in the region of interest has been added to the discussion section (Section 5.6 of the revised manuscript). We showed that the expected Joule heating rates at 110 km altitude (based on Baloukidis et al. 2023 and Günzkofer et al., 2024 (https://doi.org/10.1029/2023EA003447)) are considerably lower than the estimated particle precipitation heating rates. However, especially for the May 2024 measurements, the Joule heating rates might contribute significantly to the uplift which we added as a statement to the conclusion section.

Reviewer:

Line 189: "significantly" have you tested the result for significance. For example, a Kolmogorov-Smirnov test comparing the underlying data behind each profile with each other.

Authors:

The statement in Line 189 referred to the fact that in Figure 6 a) of the original manuscript, the yellow line deviates notably from the other two profiles (and the climatology) below about 90 km. This is because at these altitudes, the dual-frequency fit is bound to the initializing a priori profile which is different for December and May. We changed the wording in the respective sentence.

Reviewer:

Line 199: "the difference between both profiles is within the measurement uncertainty and conclusions have to be drawn carefully" is a very sensible statement but have the authors attempted any significance testing of their results.

Authors:

This statement was omitted in the revised version because we saw it prudent to apply the respective climatologies as baseline and not the low electron profile from December 2022 (due to the previously discussed data noise issue for low electron densities). We removed statements about significance as we did not apply a statistical test.

Reviewer:

Line 205: I wonder is there an independent neutral temperature measurement over EISCAT at this time? There have been temperature mappers and lidar focused on the region though I admit I am uncertain of the data range that exists. Alternatively, is there any satellite data for the interval that could be used?

Authors:

Unfortunately, the Tromso Na Lidar data is available only until 2019 online (Tromso LIDAR data). The coverage of the temperature measurements is about 85 – 100 km (though often less) and of course only at night times. We are not aware of any other available temperature measurements in this altitude range for the investigated time. We

add a plot of EISCAT ion temperature profiles (UHF manda) below. The profiles show a clear increase of temperature at 100 km altitude for the times of presumably high particle precipitation (note that below 100 km the profile is equivalent to the climatology a priori profile).

[Figure]

Reviewer:

Line 222; the jump (or rather a kink in the profile) at 110 km appears in both cases in Figure 3, it is reduced when NRLMSIS is used but it is absolutely still there. Therefore, it is erroneous to state that "This jump is only found when the standard EISCAT… not when NRLMSIS climatology profile is used". Please, correct this. I would also note that the jump is in all time sectors (fig 3a) but more pronounced in the noon and dusk sectors. The fact it persists at all times may suggest that the SNR problem is not just constrained to low density periods. In fig3b the jump is most pronounced in the dusk, then the noon and is indeed much reduced (to vanishing) in the two other sectors. This needs some thought and explanation; I suspect it may be connected to the SNR issue and how using NRLMSIS will mitigate that to a degree at higher densities, but I am unsure.

Authors:

We appreciate the comment. With the new analysis settings, the jumps/kinks are no longer present for any sector/conditions. We assume that the kinks were mainly caused by the high noise issue (in combination with the transition of a priori settings) and were less pronounced for the NRLMSIS initialized fits since the NRLMSIS profile is closer to the fitted profile than the EISCAT a priori profile at the position of the kink (100 km).

Reviewer:

Line 245: fluxes and ionization rates are often presented in units of per cm cubed, but I recommend SI units. Leaves less room for mistakes. Same for neutral densities in Figure 6.

Authors:

Done as suggested.

Reviewer:

Lines 266 and 270; essentially saying the same thing twice in quick succession.

Authors:

The second statement was deleted.

Reviewer:

Lines 264 onwards: I am not clear on how the analysis of the tidal structures leads to the conclusion on line 275. Please provide more detail of the underlying mechanism.

Authors:

Since the classical tidal theory applies pressure coordinates, the tidal equations do not explicitly express neutral density variations due to tides. However, we provided several references discussing the impact of tides on the neutral density in the lower thermosphere (Liebermann and Hays, 1994; Qian and Solomon, 2012; Truskowski et al., 2014; Maute et al., 2022; Yue et al., 2023). Additionally, we compared past measurements of the tidal impact on collision frequency/neutral density with the neutral density variations shown in Figures 4 b) and 6 b). This suggests that the potential impact of atmospheric tides is too small to cause the observed variations.

Reviewer:

Figure 6b: why not include error bars on the red line?

Authors:

The red (orange) line in Figure 6b is identical to the blue profile shown in Figure 4b where the errorbars are shown (this was also the case for the original manuscript). To keep Figure 6b comprehensible, the errorbars of the red (orange) line are not plotted again here.

---

## Author Response (AR2)

Reviewer:

The quality of the EISCAT electron density was enhanced through adjustments in the integration technique. This led to decreased noise and enhanced reliability of the calculated collision frequencies. The authors' acceptance of suggestion in my previous comment is appreciated. However, the distinction between the observational findings and inferences remains ambiguous, despite this issue being raised in the previous comment. Consequently, certain sections of the study present inferred information as if it were observationally confirmed. Additionally, some conclusions are drawn based solely on assumptions or hypotheses applied in this study, disregarding other possibilities. Many of these issues were highlighted in the previous comments. It is crucial to carefully revisit comment 3 (Feature of the vertical motion in the lower thermosphere) and comments in the second circulation (starting with "Additional issues ..."). Despite the authors' agreement in their response, there are concerns that the latest version may not have been adequately revised. Therefore, the main points are repeated below as a supplement to the comment 3.

Authors:

We appreciate that the reviewer accepts the presented mitigation strategy to reduce the impact of 'noisy' EISCAT data. We have revised the wording to emphasis the underlying assumptions. The main narrative of the manuscript is the analysis of EISCAT observations during particle precipitation and to outline potential differences and impacts compared to other studies that are already published targeting other geophysical situations. The main idea is to show the differences of the observations and applied analysis depending on geophysically different processes, which provides a pathway to more observational studies with a suit of instruments providing a more detailed and comprehensive data set or modeling studies.

Reviewer:

[Observational evidence and estimation]
This research found that particle precipitation correlated with an increase in the collision frequency. Given that collision frequency is directly related to neutral density, this finding implies a rise in density. However, it is important to note that the study does not provide concrete evidence for the underlying physical mechanism. This aspect remains speculative. The text should be revised to clearly distinguish between observed facts and hypotheses. While increased atmospheric density due to upwelling is one potential explanation for the higher neutral density, it is not the only possibility. Comment 3 introduces an isobar-related mechanism. Advection could also play a role. It is crucial to recognize that this study lacks direct evidence of atmospheric dynamics such as upwelling or uplift. Neutral density could increase through various means, including advection effects. Please consider multiple angles and have a fair scientific discussion.

Authors:

We added Section 5.5 to discuss how winds along tilted isobars can cause local changes of thermospheric density. Our discussion is based on Oyama et al., 2008 "Generation of the lower-thermospheric vertical wind estimated with the EISCAT KST radar at high latitudes during periods of moderate geomagnetic disturbance".

Reviewer:

[Conclusion under an assumption/hypothesis and its inconsistency] Generally, conclusions derived from arguments or calculations based on an assumption or hypothesis are only valid when the assumption/hypothesis itself is correct. It is crucial to assess the validity or explore alternative possibilities. Please consider this principle when revising the text. Some examples are:

- The GUISDAP analysis does not fit the ion temperature at altitudes below 100 km, leading to the assumption that particle or Joule heating does not cause a temperature enhancement. Clarify how this supposition aligns with the discussion that presumes heating occurs.

Authors:

The statement that Joule heating does not have a major impact below 100 km altitude is not based on EISCAT ion temperature measurements. The previous studies by Baloukidis et al., 2023 and Günzkofer et al., 2024 applied EISCAT ion velocity measurements to calculate electric fields and consequently Joule heating under the assumption of model conductivity profiles. The assessment (hypothesis) that at and below 100 km altitude, Joule heating rates are lower than particle precipitation heating rates is based on the estimated heating rate in Section 5.3 and the profiles shown in the mentioned references. The Pedersen conductivity reduces rapidly below the altitude of its maximum (normally around 120 km) and therefore so does the Joule heating rate. The GUISDAP ion temperature analysis and its assumptions do therefore not affect this hypothesis. We tried to clarify this in the manuscript.

Reviewer:

- Equation 3 represents the scenario where all thermal energy contributes to vertical expansion, resulting in the maximum vertical wind. The upwelling duration is determined using this calculation and assuming a distance of 6-19 km. It is important to assess the validity of these assumptions. The calculated duration should be consistent with the particle precipitation duration if the assumption is acceptable. In general, if the errors are so large that the validity of the results cannot be determined, then the analysis and discussion should be inconclusive. The text should be revised with this consideration in mind.

Authors:

We added the assessment of typical precipitation event duration from Grandin et al., 2024 which is slightly lower than the minimum estimated reaction time. We stated explicitly that the large uncertainties make it difficult to pin particle precipitation as the origin of the observed collision frequency changes despite the observed profile changes with estimated particle precipitation strength. We also added, that even if precipitation is the main driver, other processes might contribute as well. Lastly, we added a statement that the vertical wind in that Section is only an estimate and not an observation.

Reviewer:

- The discussion in Section 5.6 regarding Joule heating effects falls short of being satisfactory. The authors acknowledge that the Kp index is not an accurate indicator of localized geomagnetic activity. Furthermore, its 3-hour temporal resolution is inadequate for capturing precise temporal fluctuations. The text merely suggests that Joule heating might or might not influence the observed increase in neutral density, which is essentially a self-evident statement that offers no new insights. In essence, it remains unclear whether particle heating alone can account for the rise in neutral density. The study should conclude that the exact cause of the density increase remains undetermined.

Authors:

Calculation of Joule heating rates as done in Baloukidis et al., 2023 and Günzkofer et al., 2024 requires a different EISCAT measurement mode than applied in this study, therefore we cannot do

more than compare to these recent assessments of Joule heating in the EISCAT region. We point out in the manuscript that under regular conditions, Joule heating should be considerably lower than particle precipitation heating at and below 100 km altitude. However, we acknowledge that these studies based on Kp index might not be best suited to approximate the conditions found during the measurements presented in this paper. Hence, our conclusion is that according to common literature, Joule heating should not influence the observed changes of ion-neutral collision frequencies but we cannot conclusively determine that this holds for the present conditions. We elaborated this further in Section 5.6 and added a respective statement to the conclusion section.